# Redox-enabled electronic interrogation and feedback control of hierarchical and networked biological systems

Sally Wang[1,2,3,6], Chen-Yu Chen [1,2,3,6], John R. Rzasa[2], Chen-Yu Tsao[2,3], Jinyang Li[2,3,4], Eric VanArsdale[1,2,3,5], Eunkyoung Kim[2,3], Fauziah Rahma Zakaria [1,2,3], Gregory F. Payne [2,3] & William E. Bentley [1,2,3] ✉

Microelectronic devices can directly communicate with biology, as electronic information can be transmitted via redox reactions within biological systems. By engineering biology's native redox networks, we enable electronic interrogation and control of biological systems at several hierarchical levels: proteins, cells, and cell consortia. First, electro-biofabrication facilitates on-device biological component assembly. Then, electrode-actuated redox data transmission and redox-linked synthetic biology allows programming of enzyme activity and closed-loop electrogenetic control of cellular function. Specifically, horseradish peroxidase is assembled onto interdigitated electrodes where electrode-generated hydrogen peroxide controls its activity. *E. coli*'s stress response regulon, *oxyRS*, is rewired to enable algorithm-based feedback control of gene expression, including an eCRISPR module that switches cell-cell quorum sensing communication from one autoinducer to another—creating an electronically controlled 'bilingual' cell. Then, these disparate redox-guided devices are wirelessly connected, enabling real-time communication and user-based control. We suggest these methodologies will help us to better understand and develop sophisticated control for biology.

Communication, or the transmission of information, builds the basis of the interconnected modern world we live in today. While electronic devices freely exchange information through electron transfer, devices that connect with biological processes are rare. We suggest this stems from the disparate communication modalities of biology and electronics[1,2], their characteristic length and time scales[3], and the orthogonal assembly processes of functioning units (cells vs. CPUs)[4]. Indeed, if one were to build an 'Internet of Life' it would necessarily interact with and build on biology's many communication modalities, including the transport of molecules, ions, and electrons, the latter being those that participate in redox reaction networks (Fig. 1a). It

would be built of biofabricated structures, the designs for which naturally evolve, and are fabricated and dissimilated without impacting their surroundings. The emergence of transient electronics[5] suggests our desire to create such a living information network, as our ability to eavesdrop on, compute, and take action on biological processes will be transformative for our society. Recognizing that redox-active molecules form the basis for electron transfer in biology, we suggest that redox can also serve to connect biological systems with electronics[1,2,6–11]. That redox-active molecules participate in redox reactions at electrode surfaces suggests that electrodes, in turn, can serve as a conduit for information exchange with biological

[1]Fischell Department of Bioengineering, University of Maryland, College Park, MD, USA. [2]Fischell Institute of Biomedical Devices, University of Maryland, College Park, MD, USA. [3]Institute of Bioscience and Biotechnology Research (IBBR), University of Maryland, Rockville, MD, USA. [4]Present address: Division of Biology and Biological Engineering, California Institute of Technology, Pasadena, CA, USA. [5]Present address: National Research Council Postdoctoral Research Associate, United States Naval Research Laboratory, Washington, DC, USA. [6]These authors contributed equally: Sally Wang, Chen-Yu Chen. ✉e-mail: bentley@umd.edu

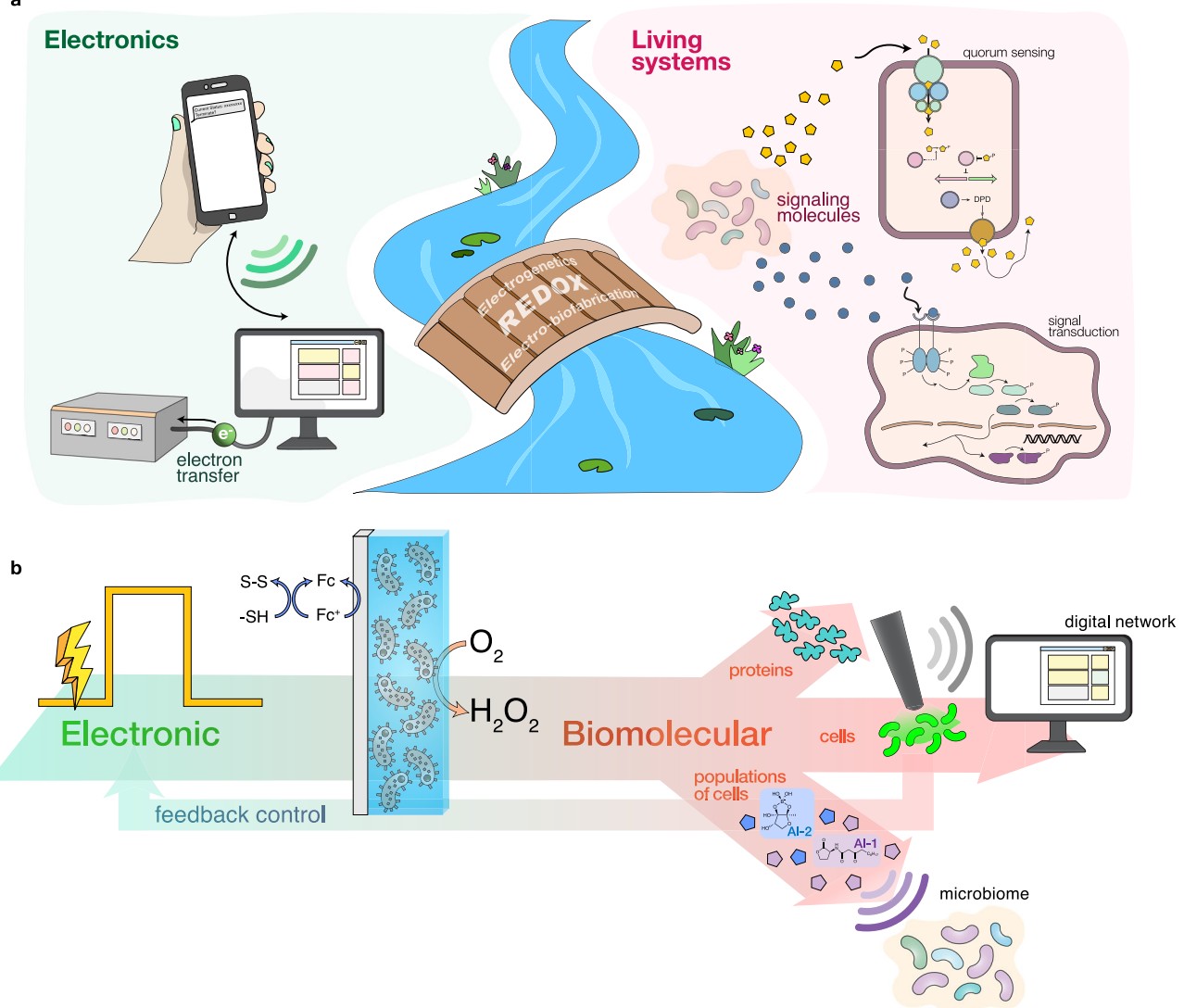

**Fig. 1 | Bridging electronic and biomolecular communication through redox.**
**a** Electronic communication (left) mainly relies on free-flowing electron transfer or electromagnetic waves for communication. Molecular communication (right), conversely, employs signaling chemical molecules for information transfer. The redox modality can connect the two disparate communication modalities with redox-active molecules that can interact with both electronics and biology. Electro-biofabrication enables the creation of transmission interfaces and electrogenetics enables specific activation of engineered genetic circuits. **b** The redox signaling modality enables connection of biology to electronics in both directions. Here, an encoded electronic input is first transduced to chemical signals: (i) an oxidized redox mediator (ferrocence, Fc) that facilitates hydrogel assembly and (ii) reduced oxygen ($O_2$ is reduced to $H_2O_2$) that is interpretable by several biological subsystems at the protein (top), cellular (middle), and multicellular (bottom) levels. Optical signals (e.g., fluorescence) generated by cells, as well as electrochemical currents (from peroxide generation), are recorded, computed upon, and fed back into process control algorithms for control, establishing an electro-bio-electro communication loop.

systems[12,13], playing a pivotal role in a wide range of biological processes, including at the protein, cellular, and multicellular levels.

In this report, we develop methods for nearly seamless information transfer between the electronics of devices and the redox-based electronics of biology. This is enabled by electronic assembly of suitable interfaces, positioning of redox-active enzymes in configurations that provide rapid information transfer, as well as specific engineering of cells that enable precisely targeted transfer of electronic information to and from electrodes. Interfaces are assembled using redox chemistry mediated by electron carriers (e.g., ferrocene) in a way that preserves stability and viability for subsequent communication with electronics via the same electrode surface. We further show the creation of CRISPR[14,15] genetic circuits that translate electronic information into biological forms, including the creation of an electronic 'language translator' shifting a population of bacterial cells to operate in a bilingual manner. We then demonstrate how electroassembled enzymes and cells can provide in real time, optoelectronic forms of biological activity that can be computed on and used for closed loop control (Fig. 1b). Specifically, we show how electroassembled horseradish peroxidase (HRP) activity can be controlled on-chip by the electronic synthesis of its substrate, hydrogen peroxide (henceforth, peroxide). We employ electrogenetics and the *oxyRS* stress regulon[16] of *Escherichia coli* to demonstrate how engineered and native cells can be electronically stimulated (again through peroxide production) and recorded (via real time fluorescence and electrochemical measurements) for feedback electronic control of gene expression (Fig. 1b). Finally, in a manner analogous to the multifaceted and interwoven nature of the Internet of Life, we show how locally regulated biological systems can be networked and even controlled via a cascaded control architecture that includes a human interface

(text messaging) that supervises model-based control actions and approves their implementation.

## Results

### Electro-biofabrication: assembly of biological components onto electrodes

Biofabrication[17] has allowed biological components to be assembled on micro-, or nano-scaled electronic devices for biosensing[18–20] or even control of biological functions[21]. Here, we enlisted electro-biofabrication[22] of bio-compatible hydrogels to facilitate the assembly of enzymes and cells onto electrodes to localize signal transfer. For enzyme assembly, we show the spatially-configurable electro-deposition of a covalently-conjugated gelatin/horseradish peroxidase (HRP)[23] hydrogel film (Fig. 2a, Supplementary Fig. 1). Its assembly builds on anodic oxidation[23] but is fabricated here with spatially confined geometries enabling area-based interrogation and control of activity (Fig. 2b (ii))[24].

For cell assembly, we took inspiration from bacterial biofilms found in nature and created 'artificial biofilms', consisting of *E. coli* entrapped in a thiolated polyethylene glycol (PEG-SH) hydrogel. Cells were mixed with a solution containing 4-armed PEG-SH monomer (50 mM) and redox mediator, ferrocene (Fc, 5 mM), to facilitate the electrode-induced oxidation of thiol groups into disulfide bonds, creating a crosslinked hydrogel[23]. When an oxidative voltage (slight variations depending on electrode material) is applied, PEG-SH crosslinks, forming a film that entraps cells within the growing matrix (Fig. 2a). In Fig. 2b (i), PEG-SH hydrogel containing SYTO 9-stained *E. coli* BL21 (OD$_{600}$ ~ 5) are electro-assembled onto on patterned gold electrodes (+0.7 V, 2 min) with various geometries and surface areas (ranging from ~13 mm$^2$ to 100 mm$^2$). The cell distribution appears homogeneous and clearly defined by the geometry of the electrode. 4–6 h after *E. coli* BL21 (they constitutively secrete quorum sensing (QS) autoinducer AI-2) were assembled onto electrodes with different surface areas (25 mm$^2$ vs 100 mm$^2$), the growth media submerging the larger electrode contained ~6 times the AI-2 activity of the smaller electrode (Fig. 2b (ii)). This result suggests preservation and proportionality of cell function with electrode area. In Fig. 2c, 'artificial biofilms' are controllably deposited with custom thickness by varying the deposition time. Here, we deposited the films containing GFP-expressing DH5α-sfGFP on an optically-transparent indium tin oxide (ITO) electrode and quantified confocal Z-stack images to estimate their thickness. Results show the film thickness was correlated with deposition time, thus providing an additional orthogonal approach for designing and controlling characteristics of this 'living' material. We would like to note that PEG-SH cell assembly, as described in "Methods", is robust and highly reproducible.

To better facilitate optical assessment of electroassembled materials and cells, we transitioned to optically clear conductive ITO electrodes. In Fig. 2d, we 3D-printed a four-well optoelectrochemical device with an ITO-coated working electrodes (WE) in a prototypical three-electrode setup. Here, each well shares the same Ag/AgCl reference electrode and Pt wire counter electrode through the salt bridge casted in the central well ("Methods"). This optoelectrochemical device provides for electronic I/O, where the same electrode for assembly is used for electrochemical detection as well as electrochemical reduction of oxygen (oxygen reduction reaction (ORR); $O_2 + 2H^+ + 2e^- \leftrightarrow H_2O_2$) forming a transmitted redox signal, peroxide[7,25]. In Fig. 2e, we found that biasing the ITO electrode with −0.8 V (Supplementary Fig. 2) produced up to ~60 µM of peroxide (in 30 min), exhibiting linear dependence with duration and consequently, charge (product of time and current). We then investigated peroxide generation in the presence of a metabolizing 'artificial biofilm' in Fig. 2f. Measured current at an applied potential reflects the extent of the ORR reaction[26]. We observed dramatically reduced current when the PEG-SH films contained respiring *E. coli*, suggesting an oxygen limitation at the electrode surface for peroxide generation.

That is, with the identically assembled biofilm, we supplemented 0.4 ft$^3$ of oxygen through a built-in gas transport tube (Supplementary Fig. 3), and an increased current was observed. In addition to demonstrating the importance of oxygen transfer to the electrode surface, this result also shows how the electrode of assembly can serve as a dynamic electrochemical sensor of the redox reaction and perhaps indirectly, the respiratory activity of the assembled cells.

Together, these results show the electro-assembly of redox-active hydrogels (gelatin and PEG-SH) can be configured with various geometries (area, thickness). Their biocompatibility also allowed both materials to either conjugate with enzymes or entrap live cells while retaining the biological component's activity. Altogether, we built a biohybrid, optoelectrochemical device consisting of a living 'artificial biofilm' and an optically clear electronic platform capable of generating peroxide as the redox signal.

### oxyRS-based electrogenetic CRISPR activation (CRISPRa)

We then enlisted the versatile CRISPR transcriptional regulation system[14] to allow activation, repression, and multiplexed transcriptional regulation of genes controlled by *E. coli*'s global oxidative stress regulon, *oxyRS*. The single guide RNA (sgRNA) was placed downstream of the *oxyS* promoter (Fig. 3a), and we monitored CRISPRa activity by the expression of *gfpmut2*. First, we confirmed in suspension cultures that both CRISPR sgRNA transcription and GFP expression were peroxide-inducible (Supplementary Fig. 4a, b). Next, we applied reducing potential (−0.8 V) to electroinduce CRISPRa in the cells electrically assembled on the ITO electrode surface (Fig. 3b). Using confocal microscopy, we found that ~70% of electroinduced cells expressed GFP, while ~35% of electro-assembled cells expressed GFP when 200 µM peroxide was introduced to the growth media above the film, and <5% of the cells expressed GFP in negative controls. Importantly, the fraction of fluorescing cells increased with charge (Fig. 3c, d; Supplementary Fig. 6).

Having shown that we can electrically induce *oxyRS*-based CRISPRa in the optoelectrochemical device, we then sought to demonstrate information routing between electronic to biological signaling modalities by initiating a specific QS communication foreign to *E. coli* (i.e., *las* 'AI-1' QS system from *Pseudomonas aeruginosa*) through peroxide-mediated CRISPR. A new reporter plasmid (pMC-lasI-LAA) was constructed by replacing *gfpmut2* with the AI-1 producer *lasI* (fused with an ssRA tag), allowing CRISPRa control of AI-1 production. Prior to electroinduction tests, CRISPRa *lasI* cells were confirmed to be peroxide-inducible in liquid culture (Supplementary Fig. 5a). Similar to the results of eCRISPRa-induced GFP which routes signals based on *soxRS* and supplemented redox mediators[14], we found that AI-1 levels were directly electrically inducible via $H_2O_2$/*oxyRS* regulon and correlated to the duration of voltage application and charge (Fig. 3e). We then further demonstrated message transfer to a second population through CRISPRa-mediated QS signal routing in a consortia setup. Specifically, QS signal (AI-1) secreted from the electrode-bound CRISPRa *lasI* cells is relayed to the planktonic AI-1 responsive population (NEB10β harboring plasmid pLasR_S129T-GFPmut3, Supplementary Fig. 5c). After receiving electroinduction, the distant planktonic AI-1 reporters exhibited ~1.5-fold increase in GFP levels compared to those in the non-induced coculture (Fig. 3f).

In summary, we demonstrated on-device, peroxide-mediated eCRISPR that allows propagation of the localized electrogenetic cue across different cell populations through the electroassembled hydrogel film, and subsequent coordination of population-wide behavior with the help of QS communication. It is important to note that because we assembled cells via entrapment in a crosslinked hydrogel film, there was no need to engineer assembly features into the assembled cells[7]. There was also no need to supplement with message-conveying redox mediators, since the signal was directly transferred from the electrode. Thus, we anticipate virtually any cell or consortia of cells can be assembled and electronically stimulated

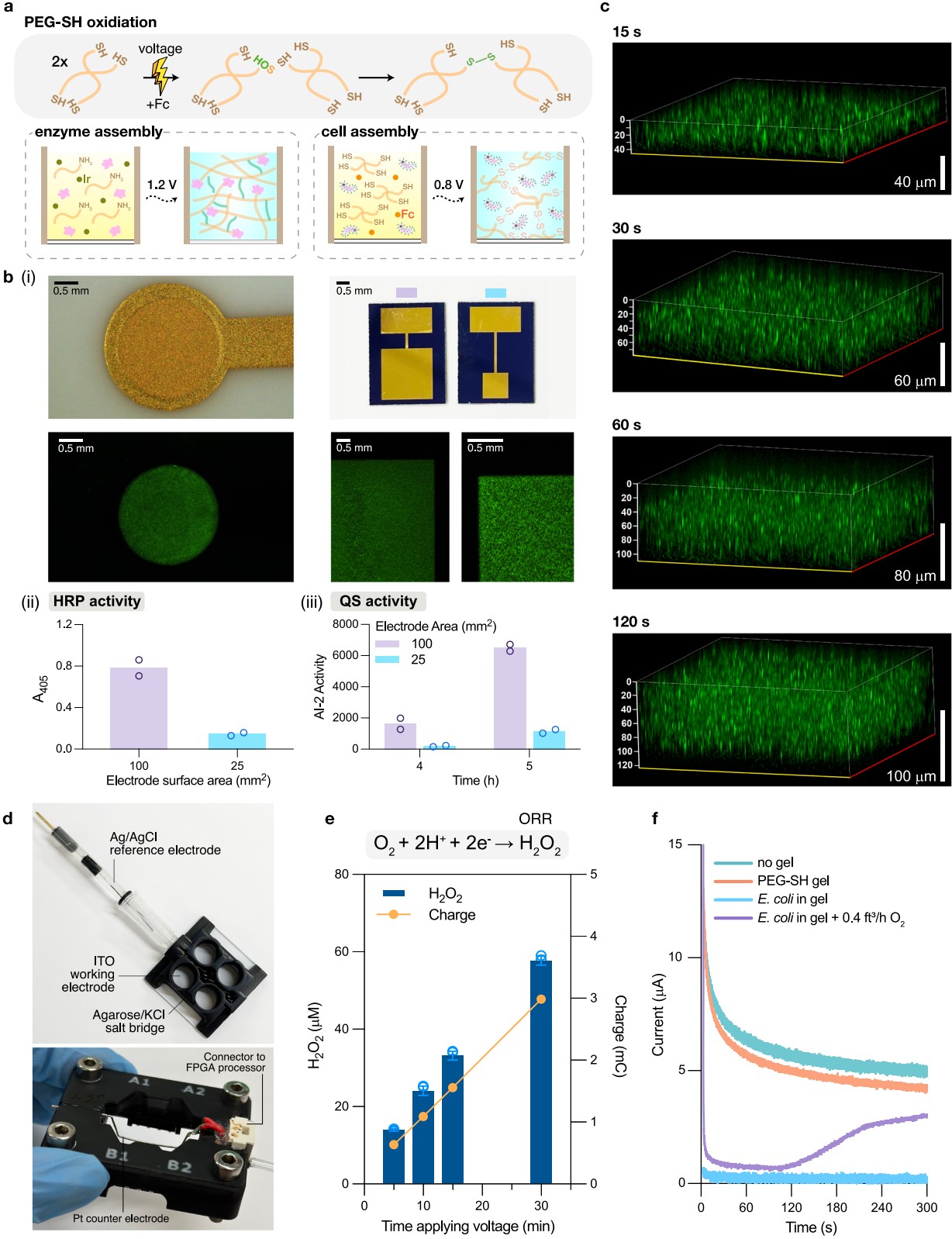

using this redox-based or other stimulus-responsive, electroassembly methodologies[22,27].

## eCRISPR inhibition and multiplexed control of QS to enable 'bilingual' communication

While we showed that enabling peroxide-mediated eCRISPR activation of QS can open a new line of communication to other microbial populations, CRISPR transcription regulation offers multiplexed function, including coincident activation and inhibition[28]. Here, we explored multiplexed control for multi-locus transcriptional regulation to enable electrically-controlled 'bilingual' QS communication. In particular, we aimed to create an engineered *E. coli* that switches its 'spoken language' (i.e., QS signaling) from its 'mother tongue' (*luxS*-based AI-2) to a 'foreign language' (*las* AI-1 from *P. aeruginosa*)

**Fig. 2 | Electro-biofabrication for assembly of biological components and integration with an optoelectrochemical electronic system. a** Schematic of PEG-SH oxidative cross-linking, assembly of HRP-conjugated gelatin hydrogel (enzyme assembly), and co-deposition of *E. coli* and PEG-SH (cell assembly). Redox mediators, Ir and Fc, facilitates the oxidation and subsequent gelation of gelatin and PEG-SH. **b** Spatially-programmable deposition. (i) Fluorescence microscopy images of 'artificial biofilms' containing SYTO-9 dyed *E. coli* BL21 assembled onto circular 2 mm-diameter, square 5 × 5 mm, and square 10 × 10 mm gold electrodes. (ii) Activity of electroassembled HRP, represented as the absorbance at 405 nm through ABTS assay. (iii) Secreted AI-2 activity from electroassembled 'artificial biofilms' containing *E. coli* BL21 cells (OD$_{600}$ = 5). Data are presented as mean (*n* = 2); open circles represent individual replicates. **c** Representative Z-stack confocal images of the 'artificial biofilms' containing GFP-expressing *E. coli* (DH5α-sfGFP). Deposition time is labeled on top of each image. X-axis (yellow), Y-axis (red), Z-axis and scale bars

(white) are shown, all units are in μm. **d** ITO-based, 3D-printed optoelectrochemical device. Clear ITO glass: working electrode and allows for optical observations. Ag/AgCl electrode: reference electrode. Pt wire (below, mounted on custom-fabricated connector): counter electrode. The three electrodes are connected through the salt bridge (1 M KCl in 1% agarose) casted in the central well of the device. **e** Peroxide (H$_2$O$_2$) generated (blue) in the ITO-based electrochemical platform as correlated to applied charge (orange). Peroxide data are presented as mean±s.d. (*n* = 4, individual replicates shown in open circles). **f** Current obtained during peroxide generation. Working electrode (WE) was poised at −0.8 V for 300 s for all samples. Teal: 20% LB. Orange: 20% LB submerging a cell-free PEG-SH film deposited on WE. Blue: 20% LB submerging an 'artificial biofilm' containing *E. coli* (OD$_{600}$ = 6) after 1.5 h of incubation at 34 °C. Purple: 0.4 ft$^3$ h$^{-1}$ of O$_2$ supplied (starting at 0 s, through built-in tubing in the connector) to 20% LB submerging an 'artificial biofilm' containing *E. coli* (OD$_{600}$ = 6) after 1.5 h of incubation at 34 °C.

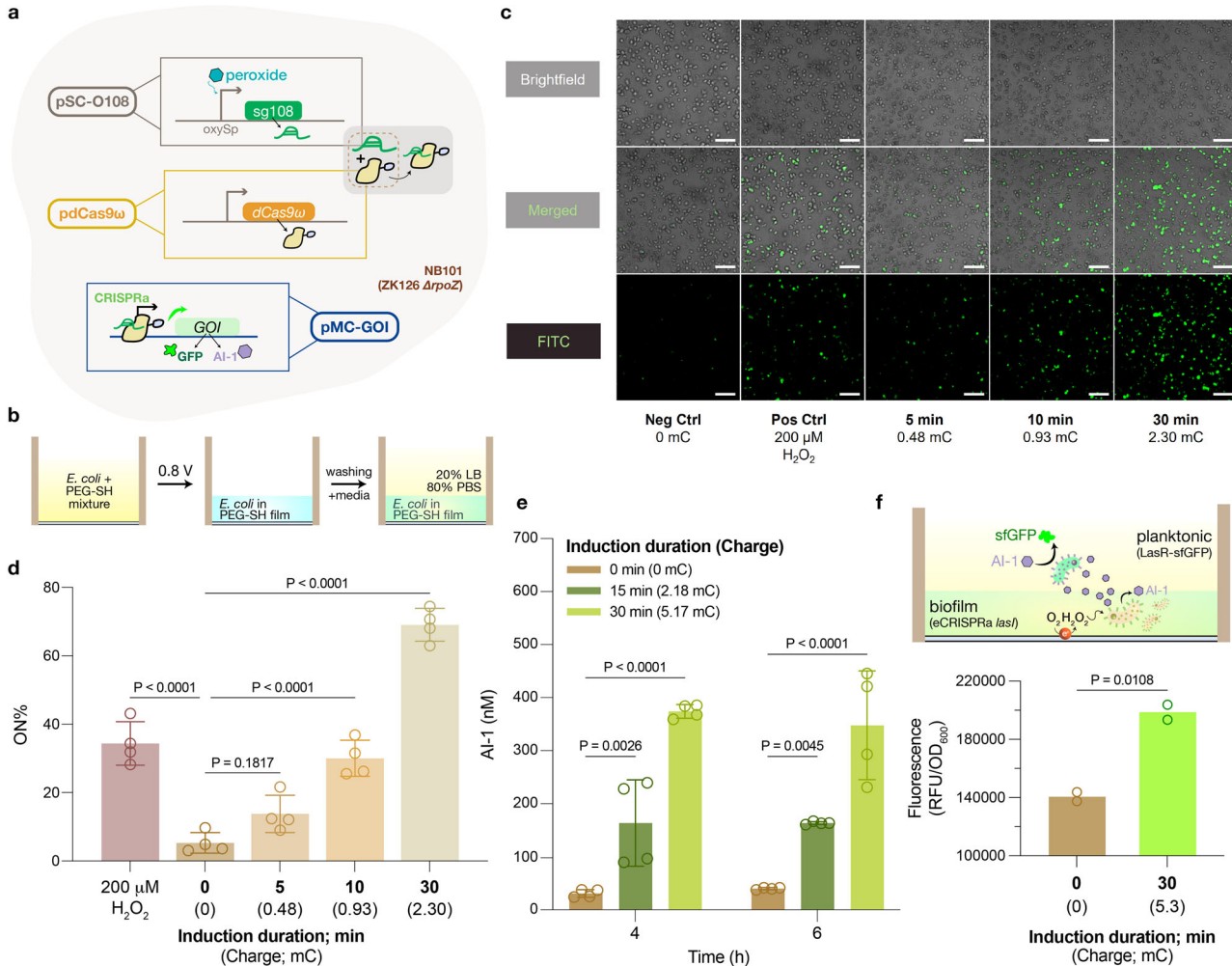

**Fig. 3 | Tunable eCRISPRa within an 'artificial biofilm'. a** Schematic illustration of the electrogenetic CRISPRa system powered by the *oxyRS* regulon. Electroinduced gRNA sg108 (from plasmid pSC-O108) forms a complex with constitutively-expressed dCas9ω to initiate transcription of the gene-of-interest (GOI) in plasmids pMC-GFPmut2 or pMC-LasI-LAA. **b** General workflow and setup for experiments with 'artificial biofilms'. After electrodeposition, we thoroughly washed the film with PBS to remove excess cell/PEG-SH solution. 150 μL of 20% LB was then added into the well to submerge the film in growth media. **c** Confocal images of *E. coli* harboring the eCRISPRa-GFP genetic cassette that were embedded in the PEG-SH film. Varying times of electroinduction and the resulting charges (millicoulomb; mC) applied to the cells are indicated below images. Neg Ctrl: negative control. 200 μM of H$_2$O$_2$ was exogenously added to culture media to activate the *oxyS* promoter as a positive control (Pos Ctrl). Top: brightfield; Middle: Merged; Bottom:

FITC filter. Scale bar = 50 μm. **d** Percentage of *E. coli* in the PEG-SH film that were activated through eCRISPRa. Data are quantified from images in Fig. 3c with ImageJ and presented as mean ± s.d. (*n* = 4). Comparisons used ordinary one-way ANOVA. **e** AI-1 assay indicating the amounts of AI-1 generated via CRISPRa *lasI* cells. Brown: 0 min electroinduction. Dark green: 15 min electroinduction. Light green: 30 min electroinduction. Data are presented as mean ± s.d. (*n* = 4). Comparisons used ordinary one-way ANOVA. **f** Measured fluorescence in a coculture comprised of eCRISPRa *lasI* cells (in film) and AI-1 fluorescent reporters (NEB10β + LasR_S129T-GFPmut3) situated in the liquid above. Data are presented as mean (*n* = 2). Comparisons used unpaired, two-sided *t*-test. In (**d**)–(**f**), *P* values were calculated between electroinduced samples and the uninduced control. Exact *P* values are provided in Source data. Individual replicates were indicated as open circles.

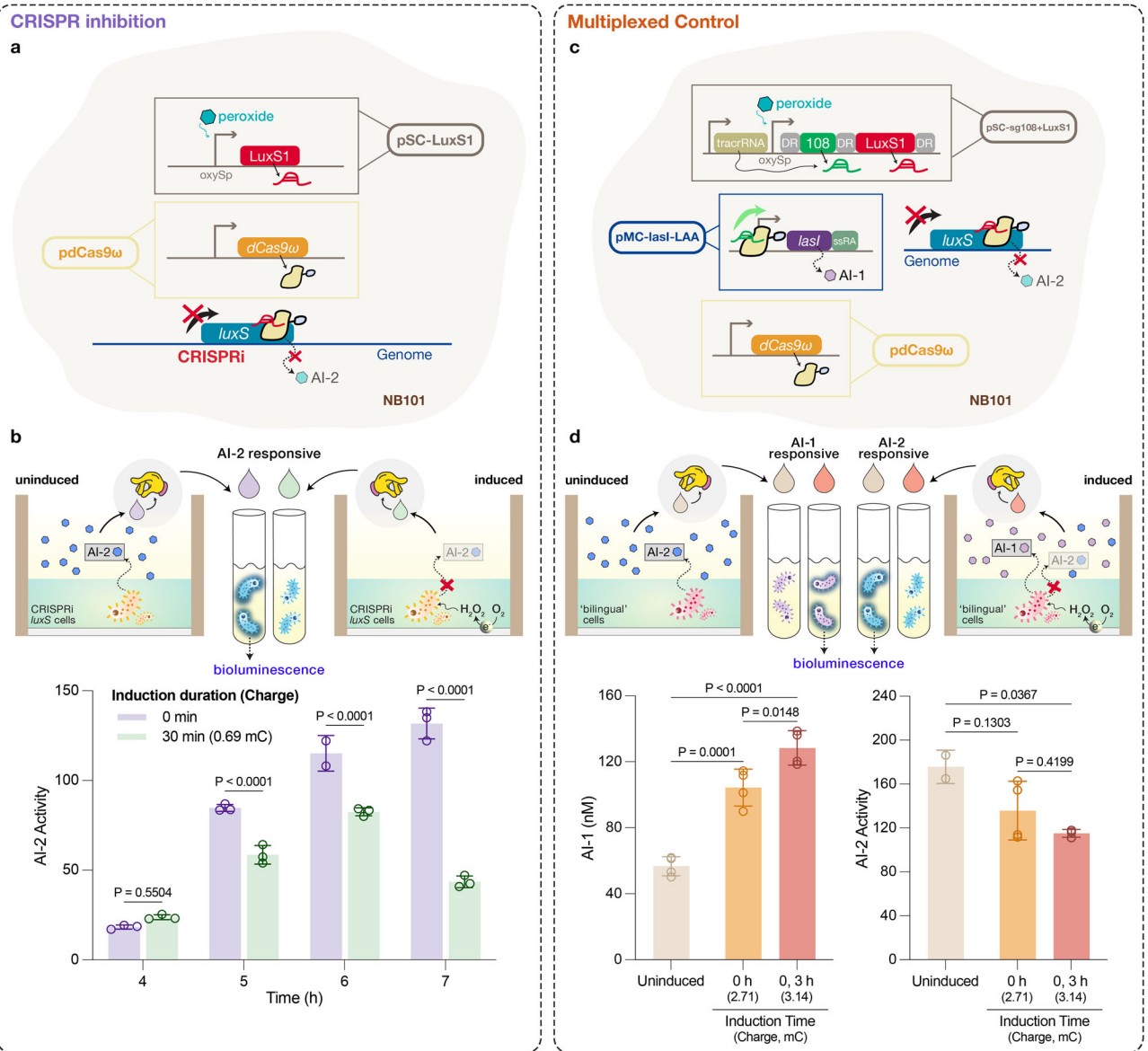

**Fig. 4 | eCRISPR inhibition of a native QS signal and multiplexed control of QS communication. a** Schematic of *luxS* eCRISPRi to inhibit AI-2 QS signaling. *luxS* specific gRNA LuxS1 is expressed under *oxyS* promoter in pSC-LuxS1. Both pSC-LuxS1 and pdCas9ω were transformed into NB101 allowing expression of gRNA and dCas9ω. **b** Measured AI-2 activity of the filtered media samples collected at various times post-induction. In-film eCRISPRi cells were electroinduced for 0 (purple) or 30 min (green) 3 h after deposition. Comparisons used two-sided mixed effect analysis. **c** Schematic of multiplexed eCRISPR for engineering a 'bilingual' strain. gRNAs sg108 and LuxS1 were flanked with the DR sequence and placed downstream of the *oxyS* promoter in pSC-sg108+LuxS1. tracrRNA was individually expressed under a synthetic constitutive promoter. Plasmids pSC-sg108+LuxS1, pdCas9ω, and pMC-lasI-LAA were all transformed into NB101 allowing multiplexed gRNA expression. **d** Measured AI-1 levels and AI-2 activity (by luminescent reporter cells) of the filtered media samples collected 7 h post-deposition. In-film 'bilingual' cells were electro-induced for 0 or 30 min at 0 or 0 and 3 h post-deposition. Comparisons used ordinary one-way ANOVA. Exact *P* values are provided in Source data. In (**b**) and (**d**), data are presented as mean ± s.d. (**b**: *n* = 3, **d**: *n* = 4). Individual replicates are indicated as open circles.

upon receiving the electronic signal. Unlike the *las* system, *luxS*/AI-2 QS system is found natively in many bacteria, including our experiment chassis *E. coli*[29]. To prohibit *E. coli* from secreting AI-2, we designed the sgRNA LuxS1 that is homologous to the AI-2 producer gene *luxS* and repurposed dCas9ω for CRISPR inhibition (CRISPRi) (Fig. 4a). The gRNA, along with a non-specific control, was expressed initially under a strong, constitutive promoter in suspension culture to examine the efficacy of LuxS1. Compared to the control gRNA, we observed a 372-fold decrease in AI-2 activity from the cultures that constitutively-expressed LuxS1, demonstrating highly effective inhibition (Supplementary Fig. 7a). Next, we replaced the constitutive promoter with the *oxyS* promoter to allow inducible expression of LuxS1. We found that AI-2 activity of the non-induced control was ~6-

fold higher than that of the induced sample, confirming peroxide-inducible inhibition of AI-2 QS system via CRISPRi (Supplementary Fig. 7b). Prior to testing electrogenetic CRISPRi, AI-2 profiles of in-film, electrode-bound cells were shown to share a similar (increasing) yet delayed trajectory with that of the suspension cultures[30] (Fig. 2, Supplementary Fig. 7c). eCRISPRi of *luxS* was subsequently found; as shown in Fig. 4b, on-electrode CRISPRi cells that were subjected to electroinduction consistently exhibited lower AI-2 activity compared to the uninduced control, proving controllable quenching of AI-2 QS through eCRISPR.

Having shown successful eCRISPRi of *luxS*, we then sought to harness the multiplexed nature of CRISPR and combine CRISPRa of *lasI* and CRISPRi of *luxS* to create a 'bilingual' strain. Though many

strategies for multiplexed gRNA expression have been previously explored[28], we chose a method that closely resembles the native CRISPR–Cas system by flanking an array of gRNAs with 'direct repeats' (DR), which are the repetitive sequences required for the tandem gRNAs to be processed by RNase III in a tracrRNA-dependent manner[31]. Based on this approach, both gRNAs sg108 and LuxS1 were flanked with DR sequences and placed downstream of the *oxyS* promoter for peroxide-inducible control (Fig. 4c). We found elevated levels of AI-1 (Supplementary Fig. 7d (i)) and reduced levels of AI-2 (Supplementary Fig. 7d (ii)) in the peroxide-induced samples compared to the uninduced control, confirming CRISPR-mediated activation and inhibition on both QS signal producers. Surprisingly, we found that the cells that experienced more than one peroxide induction displayed even lower AI-2 activity and higher AI-1 levels (Supplementary Fig. 7d). This result demonstrated dynamic control of CRISPR transcription regulation, hinting that the peroxide-mediated CRISPR activity could be 'boosted' by multiple applications of applied voltage over time to reach the desired target response. Correspondingly, dynamic multiplexed eCRISPR regulation is portrayed in Fig. 4d, in which we observed the highest AI-1 levels (2.2-fold increase) and lowest AI-2 activity (1.5-fold decrease) in the twice-induced electrode-bound bilingual cells. This again corroborated the fact that we can dynamically and electronically prompt the bilingual strain to transmit different types of signals and drive certain selected populations to elicit biological responses (e.g., bioluminescence emitted from the QS reporter cells). In sum, our results show a 'bilingual' *E. coli* strain capable of switching its QS signaling to reach different audiences based entirely on electronic cues.

## Communication with and automated control of biological activities

Through redox, 'on-off' digital states of biological systems can be assigned and programmed with conventional electrochemical instrumentation and methods. Here, we demonstrate algorithm-facilitated, automated control of both enzymatic and eCRISPR activity in protein-based and cell-based subsystems. To achieve this, we designed custom electrochemical devices and developed models/algorithms to study, communicate with, and control the behavior of these biological subsystems.

First, we redesigned the above-noted 3D-printed optoelectrochemical device for encoding and decoding enzyme activity, by regulating the activity of HRP through controlling the production of its substrate peroxide. We use electronics to 'write' chemical information that is 'stored' and then at a later time, electronically retrieved, enabling an electro-chemo-electro analog to the well-known random-access memory (RAM) of electronics. The custom electro-biochemical platform consists of a patterned ITO electrode that is separated into two interdigitated working electrodes and attached to a custom 3D-printed housing (Fig. 5a, b). Specifically, working electrode 1 (WE1) is tasked with generating peroxide (through ORR), hence 'writing data'. Working electrode 2 (WE2), on which we electroassembled an HRP-conjugated gelatin hydrogel, is responsible for 'reading' the generated peroxide and 'recording' this information (in the form of electric current) (Fig. 5b). The recording function is enabled by measuring the current generated by the HRP-catalyzed enzymatic reaction with its substrate peroxide, and the subsequent redox cycling with mediator Fc (Fig. 5c)[20,23]. We found the endpoint current (recorded at 120 s) from WE2 increased each time a reducing charge was applied on WE1 (Fig. 5d), even with long periods (60 min) in between (Fig. 5e, Supplementary Fig. 8), confirming the platform's ability to write, store, and retrieve data on enzyme activity.

To realize automated control of *oxyRS*-based eCRISPR, we first studied the dynamic induction behavior of the *oxyRS* regulon when induced electrochemically. A simpler construct stripped of the CRISPR components but retaining its ability to respond to peroxide (encoded by *oxyS* promoter) was created. In suspension culture, we performed batch experiments and repeatedly induced with bolus additions of 50 or 100 mM peroxide. During both induction periods, we observed initial increases in GFP expression that were diminished after 45 min (Supplementary Fig. 9a), presumably due to transient depletion of peroxide[32]. We also show how cell growth was largely unaffected by repeated inductions (Supplementary Fig. 9b). These investigations exhibited that repeated induction could be sustained with minimal deleterious effect on cell proliferation and subsequent gene expression[21,33].

Next, because optical (i.e., fluorescence) and electrochemical outputs either indirectly (optical) or directly (electrochemical) allow for real-time quantification that can be acted upon including through a process control scheme, they are both employed here as the primary signal modalities for bio-to-electronics communication. That is, to create a network for fully automated control, we constructed 'BioSpark', a complete bioelectronic system consisting of (i) a fluorescence detection module for gene expression measurements, (ii) a multi-channel, field-programmable gate array (FPGA)-based potentiostat for sending and receiving electronic commands (and electrochemical detection), (iii) custom graphical user interface (GUI) program for system control and enabling local and wireless communication, and (iv) a custom 3D-printed environmental chamber for humidified, temperature-controlled, and oxygenated cell culture (Fig. 6a, Supplementary Figs. 10, 11, and "Methods"). With BioSpark, multiple studies were carried out to examine the expression dynamics driven by on-device electroinduction. First, constitutive GFP-expressing *E. coli* helped us track its cell growth when assembled as 'artificial biofilms' (Supplementary Fig. 12a). Using fluorescence levels as a surrogate for cell number, we observed a 'lag-phase' of ~1.25 h immediately after electroassembly where no increase in fluorescence was detected. This lag period was consistently observed across many experiments, henceforth, we chose to initiate experiments, including electroinduction, with 1.25 h set as time zero. Next, NB101 cells harboring the peroxide-reporting plasmid (pOxy-sfGFP) were electrically induced (−0.8 V) for different durations. Similar characteristics were observed to that of suspension cultures: (i) we saw an initial surge in electroinduced gene expression followed by a deceleration to a plateau (Supplementary Fig. 12b); (ii) this system retained its inducibility and repeatability, as shown by the increasing GFP levels when higher charge was placed (Supplementary Fig. 12c) and over multiple times (Supplementary Fig. 13). Based on these findings, we developed a phenomenologically-derived mathematical model of *oxyRS*-based electrogenetic expression ("Methods") so that a custom control algorithm ("Methods") could be implemented.

We then electroassembled the electrogenetic cells onto BioSpark's electrodes and implemented a custom control algorithm, demonstrating for the first time, fully electronic feedback control of gene expression. Figure 6b illustrates the automation workflow: gene expression, as represented by GFP levels, was measured and digitized via the fluorescence detection module to be relayed to the control algorithm. Based on many heuristic observations, we defined three model parameters that provide real time insight regarding the prevailing expression profile: (i) a slope ratio that represented the change in the increase of fluorescence level, (ii) a threshold value for implementing simple proportional control, and (iii) a target value that served as an objective function ("Methods"). That is, we calculated in real time, the prevailing slope of the recorded fluorescence values. The maximum slope was updated continuously. We then computed the ratio between the prevailing and the maximum slope. If two consecutive ratios fell below the user-defined threshold, an additional potential was provided, and for consistency, each sequential pulse was of identical charge. Importantly, peroxide production is quantified by the prevailing current generated from the provided potential. These actions were repeated until later, when the target value was reached and the experiment was terminated (Supplementary Fig. 12d). With this simple control scheme in place, we then tested its efficacy. We

**Automated electrical control of enzyme activity**

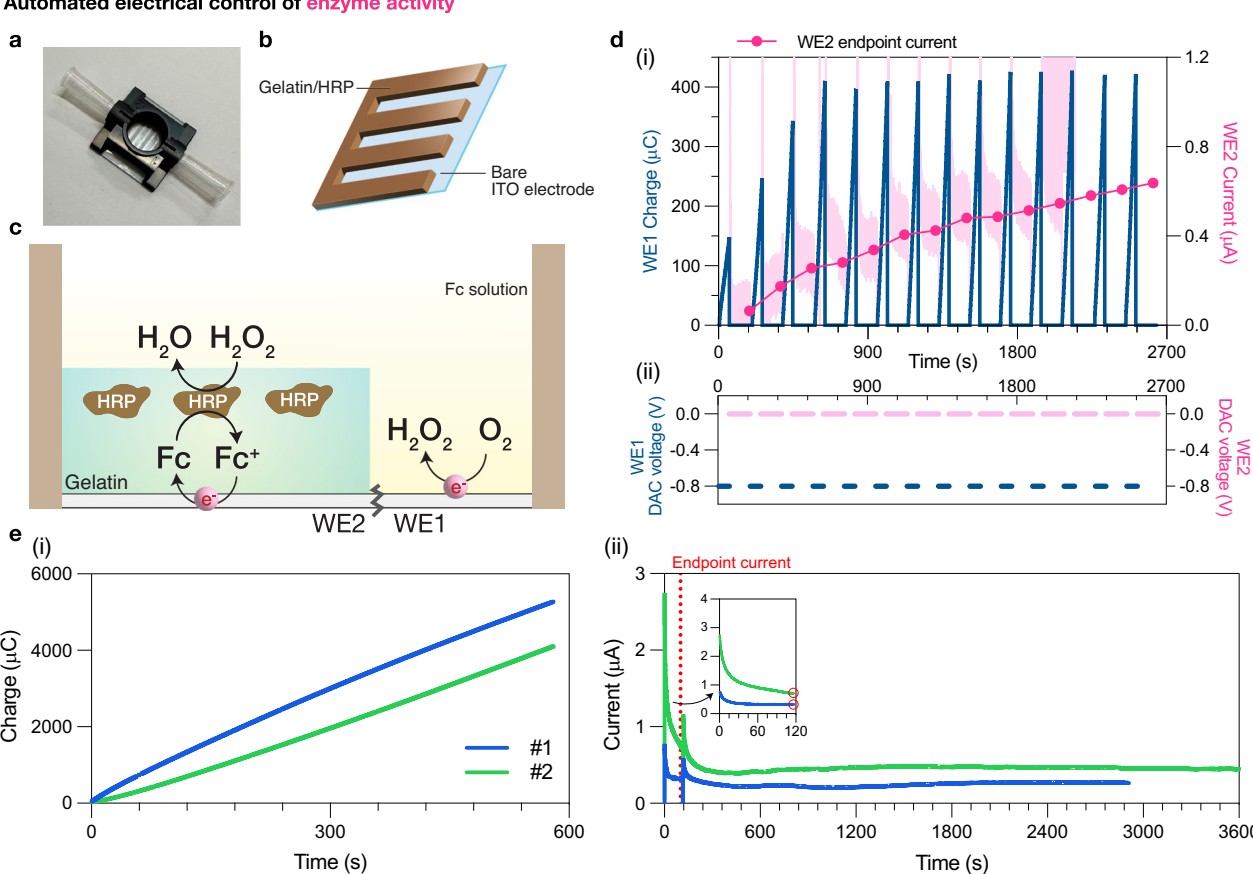

**Fig. 5 | Automated dynamic control of enzyme activity. a** Custom electrochemical platform with a patterned ITO glass slide attached to a 3D-printed housing. **b** An interdigitated electrode is generated through laser-cutting the ITO-coated glass into a zig-zag pattern to ensure the generated peroxide (on the bare electrode) is at the vicinity of the HRP/gelatin hydrogel for detection. **c** Working mechanism of the electro-biochemical platform. Horseradish peroxidase (HRP)-conjugated gelatin is pre-deposited on working electrode 2 (WE2). Working electrode 1 (WE1) consists of a bare ITO electrode and is tasked with generating peroxide by applying a reducing voltage (−0.8 V). In-gel HRP (WE2) then catalyzes the generated peroxide, and the resulting electron cycles with Fc that is present in the

solution above. A current (from the cycling of Fc) can be read when an oxidizing voltage is applied (0 V vs Ag/AgCl). **d** Demonstration of data writing and storage. (i) Generated charge on WE1 (navy) and current obtained from WE2 (pink) were plotted over time. Pink filled circles: endpoint current recorded at 120 s. (ii) Voltages applied on WE1 and WE2 over time. **e** Demonstration of 'long-term' data storage. (i) Total charge applied on WE1 for 600 s. (ii) The following currents obtained on WE2 (inset: 0–120 s). Red line and open circles indicate when the endpoint current was recorded (at 120 s). Blue: first round (#1) of data writing (600 s) and recording (-3000 s). Green: second round (#2) of data writing (600 s) and recording (3600 s), conducted immediately after the first round.

assembled both peroxide reporters (Supplementary Fig. 13) and eCRISPRa cells (Fig. 6c) into 'artificial biofilms' and controlled their expression with the automated bioelectronic system. Simple visual examination of the profiles (Fig. 6c (i)) shows how electroinduced cells rapidly increased their fluorescence immediately following the applied charge (within 15 min). Commensurate with both degraded signal molecule (peroxide[32]) and potentially reduced metabolic function[34], the GFP fluorescence slowed and reached a plateau after an applied charge. In all cases (Fig. 6c (ii)), our algorithm effectively identified each plateau region in the GFP expression profile and at each time signaled the potentiostat to apply a new charge (2 mC) enabling extra and consistent electroinduction[35]. Then, in Fig. 6c, cells ultimately reached the target expression level ($3.2 \times 10^6$ AU) after four electroinductions (total 8 mC). Coincidently, in Supplementary Figs. 14 and 16 we depict the current associated with electroinduction. As noted above, this electrochemical data dynamically reveals the rate of peroxide generation, which in turn, reflects the available dissolved oxygen at electrode surface. Given constant input of oxygen in a well-mixed system (Supplementary Fig. 3), this also indirectly indicates the respiratory activity of the cells immobilized in the hydrogel film. That the current was always above zero and monotonically decreased in time suggests both that the cells had oxygen and their continued

growth in the films progressively inhibited its transfer to the electrode. In sum, these data demonstrate for the first time, closed-loop direct electronic control of gene expression on both direct peroxide-induced *oxyRS* expression as well as CRISPR-mediated *oxyRS* targeting[36] enabled by the bioelectronic BioSpark system.

## Network integration for cascaded and supervised control

Biological systems, including living systems, are autonomously controlled locally as well as in cascaded networks via biological feedback (e.g., enzyme pathways with end product inhibition[37], microbiome composition[38,39], central and peripheral nervous systems[40], the innate and adaptive immune systems[41,42], including among seemingly disparate systems like the gut brain axis[43]). Owing to our ability to electronically survey, compute, and control biological systems, in Fig. 7 we show how the above biological subsystems are used to create a network allowing multidirectional communication and cascaded, supervised feedback control of dynamic 'living' subsystems. The full network (as illustrated by the system diagram in Supplementary Fig. 15) is built upon the communication between (i) the BioSpark system for electrogenetic control of device-localized eCRISPR activity in cells, (ii) the bio-electrochemical platform for electric control of enzyme activity, and (iii) human users through the internet and text

**Automated electrical control of eCRISPR-mediated expression**

**Fig. 6 | Automated dynamic control of gene expression. a** Schematic illustration of the BioSpark system allowing electrochemical induction and real-time fluorescence/electrochemical measurements. **b** Automation experiment workflow. 'Artificial biofilms' containing engineered bacteria are assembled onto the optoelectrochemical device. The gene expression level, represented by the emitted fluorescence, is constantly monitored by BioSpark and sent to the PC (diamond) for processing. Our custom algorithm (Supplementary Fig. 13d) then determines the status of expression via calculating the rate of fluorescence increase ("Methods"). When expression begins to tail, a user-specified threshold is met, and the potentiostat triggers an induction voltage for a specified charge. The experiment is automatically terminated when the fluorescence level is above a user-defined GFP threshold. **c** Automated dynamic control of eCRISPRa-regulated gene expression. (i) Fluorescence level within the 'artificial biofilm' containing eCRISPRa bacteria (NB101 harboring pSC-O108, pdCas9ω, and pMC-GFP). Fluorescence measurements were taken every 15 min (0.25 h). Experiment was terminated automatically

when the mean fluorescence was above the specified target (blue dotted line). Brown (open squares and line): negative controls to which no induction voltage was applied. Yellow zones indicate the durations when the induction voltage (−0.8 V) was applied to the experimental samples, and 0.4 ft$^3$ h$^{-1}$ of oxygen was supplemented to both the experimental and negative control samples. A total charge of 2 mC was applied in each yellow zone. $N = 2$ individual replicates are shown in open circles and open squares. For each biological replicate, three fluorescence measurements were performed and averaged. Lines (with no symbols) indicate the mean of biological replicates. (ii) Ratio of slope, S, to $S_{max}$ computed by our custom algorithm. Ratio threshold (purple dotted line) was set at 0.4. Orange line: algorithm applied the initial induction voltage. Yellow lines: ratio threshold met thus another electroinduction was triggered. Teal dotted line: both the ratio threshold and the induction level target were met; hence no voltage was applied, and the experiment was terminated. Filled circles represent the mean of individual replicates ($n = 2$).

messaging. A simple algorithm was developed to control the electro-biochemical platform (denoted 'remote' system), which was then linked the 'local' electrogenetics system (BioSpark) for cascaded control of eCRISPR within the local BioSpark. As shown in Fig. 7a, local GFP expression levels of eCRISPR cells were monitored via BioSpark and these were relayed to the independently controlled bio-electrochemical platform for regulating HRP activity at a remote location. When the BioSpark algorithm detected a plateau in the GFP expression profile (again defined by the user-set ratio threshold ("Methods"), a message was wirelessly conveyed to the remote bio-electrochemical platform over the internet. While there is no direct functional link between HRP activity and electrically-induced GFP expression in cells, the HRP serves here as a chemical/biological memory that is electronically addressed and in essence serves as an 'actuation checkpoint'. Specifically, information pertaining to HRP's enzymatic activity, as represented by the output current from the remote platform, is purposely designated to control the local BioSpark. In Fig. 7b, after the message from BioSpark was conveyed to the HRP electro-biochemical platform, its programmed procedure including the production of hydrogen peroxide (−0.8 V for 600 s on WE1) and retrieval of its level (0 V for 120 s on WE2) was then initiated.

The resulting output current was then compared to another user-defined parameter (current threshold) that is used to feedback on and ultimately terminate eCRISPR activity at the local BioSpark. Initially at low peroxide level/HRP activity (Fig. 7b), this process served to cue the local BioSpark for additional electroinduction. Once the HRP activity had surpassed the threshold, the actuation checkpoint triggered a termination process for the BioSpark. This included both text messaging verification from human users on mobile phones and, once approved, the subsequent photobleaching of the electroinduced GFP (Fig. 7b and Supplementary Fig. 17). In Fig. 7b, the photobleached reduction in fluorescence was observed. This demonstration, while there is no readily apparent biological basis for linkage (as is the case for feedback-controlled enzymatic pathways or innate/adaptive immunity), its example portends the future, where seemingly disparate biological systems can be studied, perhaps eventually linked and then even controlled.

## Discussion

In hopes of building an interconnected, feedback-controlled network of biological systems, we initially applied electro-biofabrication methods to assemble biological components in a manner that

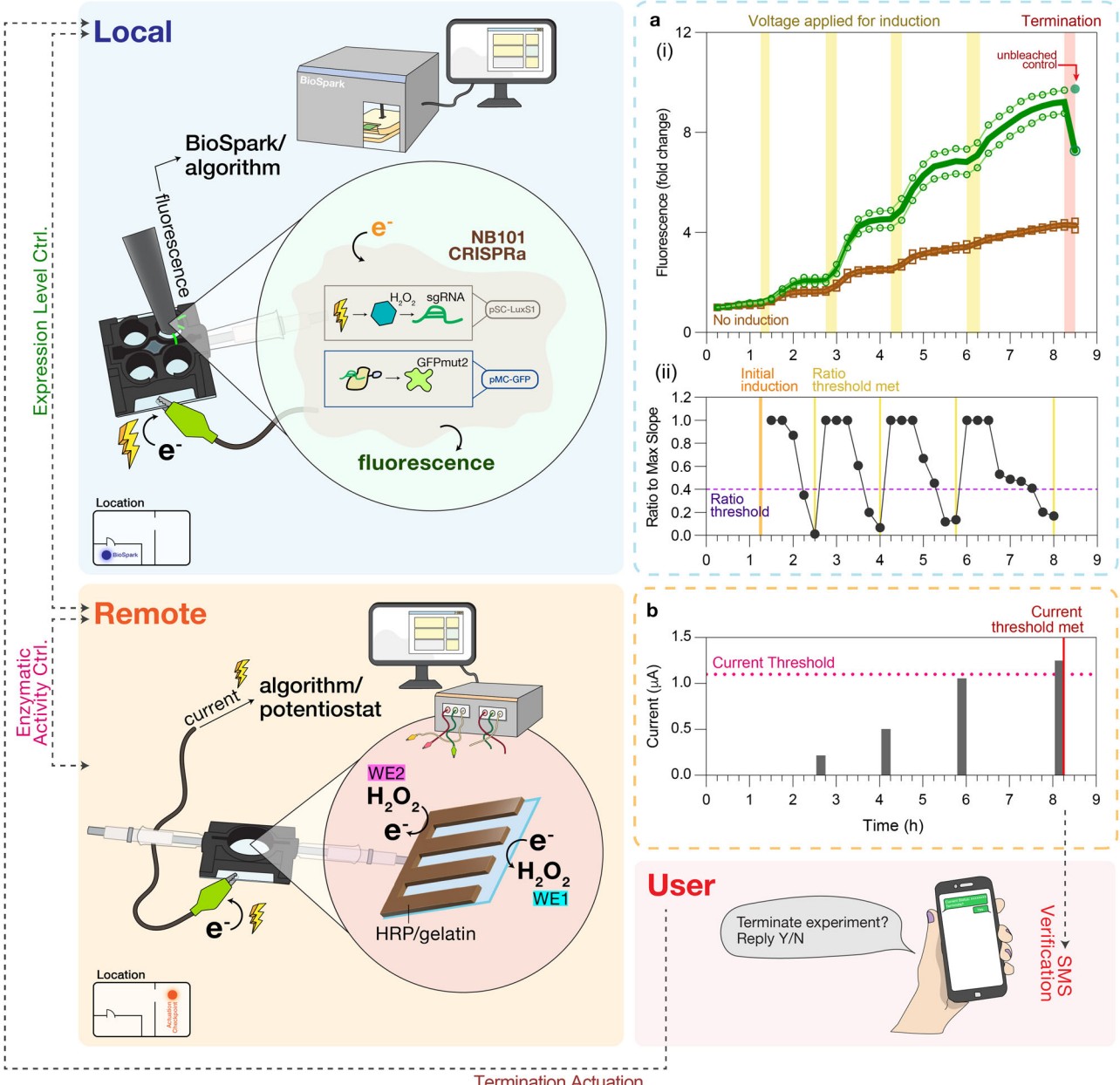

**Fig. 7 | Network integration to enable remote feedback control of eCRISPR activity within the 'Internet of Life'. a** Automated feedback control of eCRISPRa-regulated gene expression (i) Fluorescence level of the 'artificial biofilm' containing engineered eCRISPRa bacteria (NB101 harboring pSC-O108, pdCas9ω, and pMC-GFP; analogous to Fig. 6). Fluorescence measurements were taken every 15 min (0.25 h). Brown (open squares and line): negative control to which no induction voltage was applied. Yellow zones indicate the durations when the induction voltage (−0.8 V) was applied to the experimental samples, and 0.4 ft³ h⁻¹ of oxygen was supplemented to both the experimental and negative control samples. A total charge of 2 mC was applied during each yellow zone. The red zone indicates when the selected individual sample was being photobleached. $N = 2$ individual replicates

are shown in open circles and open squares. The green filled circle represents the unbleached control after termination. For each biological replicate, three fluorescence measurements were performed and averaged. Lines (with no symbols) indicate the mean of biological replicates. (ii) Ratio of slope, S, to $S_{max}$ computed by our custom algorithm. Ratio threshold (purple dotted line) was set at 0.4. Orange line: algorithm applied the initial induction voltage. Yellow lines: ratio threshold met and proceeded to trigger the remote 'actuation checkpoint'. **b** Current threshold (pink dotted line) was set at 1.1 μA. The red line indicates when the current threshold was met, which prompted the PC at the remote location to send a text message to users. Filled circles represent the mean of individual replicates ($n = 2$).

enables information exchange through redox. Enzymes and cells were placed in direct proximity to electrodes, with the latter forming 'artificial biofilms', a biomimetic structure. In this way, electro-biofabrication ensures efficient information flow at the bioelectronic interface and minimizes transport-limited heterogeneous responses of suspended biological systems[35,44]. We would like to note that similar to the naturally-occurring biofilms, electrodeposited artificial biofilm is a three-dimensional structure comprising the entrapped bacteria. While

it is well-known that the biofilm-dwelling bacteria undergo phenotypic shifts and adopt different lifestyles relative to their planktonic selves[45], further studies can help to characterize physiological changes in entrapped bacteria, potentially opening new vantagepoints for electrogenetic control. That is, electroassembly opens avenues for immobilizing a multitude of living organisms, as our approach does not require extra protein or genetic engineering for assembly. This also relieves any extra metabolic burden associated with attachment and

provides more effective use of resources for programmed functions[46]. Moreover, our assembly approaches are scalable and spatially programmable, inviting future opportunities for enzyme/cell grafting on complex formats like three-dimensional, miniaturized, arrayed electrodes; or on soft, flexible, bio-compatible materials, for wearable, ingestible, or other portable systems[47,48].

We then engineered *oxyRS*-based, peroxide-mediated eCRISPR that greatly expands the emerging repertoire of electrogenetics[7,14,49]. eCRISPR enables both upregulation and inhibition in a multiplexed manner for control of a variety of genetic targets. Here, we show that electronically programming two QS-related functions enables broad utility for manipulating microbial communities. Widespread in nature, QS signaling represents an ideal candidate for dispatching a highly-localized electrochemical cue to other ex situ biological populations with none or minimal genetic rewiring[29,50,51], and achieving coordination within a natural or synthetic consortium[8,52]. Despite the many advantages peroxide brings as an actionable bridge to connect biology and electronics, its cytotoxicity and ability to induce mutations warrant careful design[53]. Fortunately, electroinduction via $H_2O_2$ generation can be precisely controlled (Fig. 2e) and can be limited to a non-toxic yet above-threshold level. Additionally, in our previous work, we employed CRISPRi to downregulate defense response genes enabling more focused redox signaling via P*soxS*[14]. We anticipate additional multiplexed *oxyRS*-based CRISPR tuning, and the use of different hosts may enable more widespread use of electrode-generated hydrogen peroxide.

Next, automated electronic control of both enzymatic and genetic activities were demonstrated through device and algorithm development. While we have previously shown on-chip electronic modulation of chimeric proteins to control enzymatic activity during the generation of QS signals[21], here we focused on the versatile redox-active enzyme, HRP. On top of electrochemical biosensors[54], HRP is employed in many traditional, usually optical, biochemical assays[20,55–57]. Our current platform, with simple modification, could bring forth novel electrochemical procedures or even modular 'bioproduction breadboards' to interrogate or control biological processes[47,58,59], especially considering that there is an extensive collection of redox-controlled enzymes. To realize electronic control of gene expression, responses from genetic activity were assessed using both light (GFP) and current ($H_2O_2$ generation) facilitating real time on-line control. We suggest that direct, electronic assessment and control augments the many transformative technologies associated with optogenetics[60–62] and magnetogenetics[63]. By tapping into and controlling the native redox networks that are ubiquitous in biology, our approach offers a vast trove of targets and eventual applications. For example, eCRISPR provides a readily adaptable platform for simultaneous control of various genes and proteins via multiplexed transcriptional regulation. These include the production of redox-active molecules[35,64,65] that can be electrochemically assessed and digitized by our multimodal device and platform, establishing yet another avenue for communication from biology to electronics. We further envision data analytics approaches such as machine learning, that when coupled with artificial intelligence, will greatly enhance phenomenological and first principles methodologies for optimized electrogenetic control.

Finally, in this work, we demonstrate wireless integration of bioelectronic systems forming a networked electrogenetic system. Our real-time, feedback control of eCRISPR activity was performed entirely electronically via redox signals and wireless internet connection/communication, suggesting convenient monitoring and feedback control of cells inside bioreactors for smart biomanufacturing[66]. Ultimately, all enzymatic and electrogenetics components in this study (that is, redox/peroxide signaling, CRISPR techniques, and QS communication) are either native to or can be easily ported to many biological systems; hence, we foresee interconnected biohybrid networks that provide the foundation of a variety of intelligent systems and devices[67]. These include self-regulated, biomedical devices for in situ

production and delivery of a therapeutic[68,69], 'smart' agricultural systems for rhizosphere manipulation[70], or 'sense-and-clean' strategies for battling environmental pollution[18,71]. Taken in sum, the concepts demonstrated here may serve as a blueprint for a more connected world in the future.

## Methods
### Chemicals
Potassium chloride (KCl), $H_2O_2$ (30%), phosphate-buffered saline (PBS), potassium phosphate monobasic ($KH_2PO_4$), potassium phosphate dibasic ($K_2HPO_4$), potassium hexachloroiridate(III) ($K_3IrCl_6$, Ir), and gelatin from porcine skin (gel strength ≈ 175 g Bloom, Type A, G2625) were purchased from Millipore-Sigma. All antibiotics (ampicillin, kanamycin, and chloramphenicol) were purchased from Millipore-Sigma. Lysogeny broth (LB) and agarose were from Fisher Scientific. Fluorescent beads and Pierce® horseradish peroxidase (HRP) were purchased from Thermo Fisher Scientific. 4-arm PEG-SH (MW 5000) were from JenKem USA. 1,1'-Ferrocenedimenthanol (Fc) was from Santa Cruz Biology. AI-1 (N-3-oxo-dodecanoyl-L-homoserine lactone) was from Cayman Chemicals. Fc and AI-1 were initially prepared as 1 M and 5 mM DMSO stocks.

### Culture media and conditions
Unless otherwise indicated, cells were grown overnight in LB at 37 °C, 250 r.p.m. shaking, inoculated at $OD_{600} = 0.1$ in LB media the following day, and grown until the indicated cell density (optical density at 600 nm, $OD_{600}$). Optical density was measured using an UV-Vis Spectrophotometer (Beckman Coulter).

### Plasmid construction
All bacterial strains and constructs used in this study were listed in Supplementary Table 1. All enzymes, competent cells and reagents were from New England Biolabs and used according to provided protocols. Q5 polymerase and primers in Supplementary Table 2 (relevant gene sequences in Supplementary Data 1) were used for PCR reactions. DpnI digestion, polynucleotide kinase phosphorylation, T4 ligations, Gibson assembly, and *E. coli* chemical transformation were performed using New England Biolabs product protocols. DNA clean-up (Zymo Research), gel extraction (Zymo Research) and plasmid preparation kits (Qiagen) were performed using provided protocols. Synthetic gene fragment containing the multiplex crRNAs was purchased from Thermo Fisher Scientific. Synthetic gene fragment containing the 108 spacer and gRNA scaffold was purchased from Integrated DNA Technologies (IDT). Synthetic gene fragment containing tracrRNA was purchased from Integrated DNA Technologies (IDT).

Plasmid pSC-O108 for peroxide-inducible expression of sgRNA sg108 was made via PCR amplification and Gibson assembly. First, pSC-108gRNA[14] was removed of *soxR*, *soxRS* promoter region, and sgRNA sg108 by restriction digestion with ClaI and BamHI. Gene fragment containing *oxyR* and *oxyS* promoter was amplified from plasmid pOxy-LacZlaa[7] with primers SW01 and SW02. After DpnI treatment, fragments were ligated by Gibson Assembly. Next, gene fragment containing spacer 108 and gRNA scaffold was then PCR amplified with primers SW03 and SW04 and inserted into the linearized intermediate product via primers SW05 and SW06 to generate the construct pSC-O108.

Plasmid pSC-LuxS1 for peroxide-inducible expression of sgRNA LuxS1 was made via site-directed mutagenesis. Spacer 108 in plasmid pSC-O108 was swapped with LuxS1 using primers SW07 and SW08. Plasmid pSC-sg108+LuxS1 for peroxide-inducible expression of crRNAs spacer 108 and LuxS1 was made vis PCR linearization, restriction digestion, and T4 ligation. pSC-O108 was initially PCR linearized using primers SW09 and SW10 to remove the spacer 108 and the gRNA scaffold, as well as adding restriction sites BamHI and XhoI. Gene fragment containing crRNAs spacer 108 and LuxS1 was inserted into linearized pSC-O108 backbone via restriction digestion and T4

ligation. After this, the gene fragment containing the tracrRNA was inserted into the product generated from the previous step via restriction digestion and T4 ligation to generate the final construct.

Plasmid pLuxS1 for constitutive expression of sgRNA LuxS1 was constructed via site-directed mutagenesis using primers SW11 and SW12 to swap the spacer 108 in plasmid pS108gRNA[14] to LuxS1. Plasmid pMC-lasI-LAA was constructed via PCR amplification and Gibson assembly. pMC-GFP[14] was linearized and removed of GFPmut2 by primers SW13 and SW14. Primers SW15 and SW16 were used to amplify and add the LAA ssRA tag to lasI. lasI with the added LAA tag was then inserted to the linearized backbone via Gibson assembly. Plasmid pOxy-sfGFP-AAV was constructed via site-directed mutagenesis using primers SW17 and SW18 to add the AAV ssRA tag to pOxy-sfGFP[44].

### Electrodeposition of *E. coli* and PEG-SH

*E. coli* grown to mid-log phase were harvested via centrifugation at $3000 \times g$ for 10 min, then resuspended in 1× PBS to 2× the desired OD. To prepare the 2× PEG-SH solution, 100 mg/mL of PEG-SH were dissolved in phosphate buffer (PB, 0.1 M pH = 7.4) containing 10 mM Fc, as described previously by Li et al.[23]. Prior to electrodeposition, the two solutions were mixed at a 1:1 ratio. We then performed chronoamperometry, poised at 0.8 V for 30 s (or stated otherwise), to initiate oxidation of the thiol group for cross-linking. The endpoint charge was recorded for each run. Excess cell/PEG-SH solution was then removed, and the generated film was carefully washed via gentle pipetting with PBS to remove any unbound cell/PEG-SH.

### Electrode chip fabrication

Gold-patterned electrodes were purchased from Platypus Technologies and Pine Research. Steps for fabricating the custom patterned gold electrode on silicon wafers by Platypus Technologies were as follows: First, metal deposition was performed on standard 4-inch silicon wafers using a Denton thermal evaporator (Denton Vacuum), with metal deposition rates of $2–3 Å s^{-1}$. Specifically, a 50 nm chromium adhesion layer was evaporated, followed by 100 nm gold. Next, photolithography utilized direct writing of photoresist via a DWL66fs laser writer (Heidelberg Instruments), guided by a laser exposure map designed in AutoCAD (Autodesk). Photoresist spin-coating and development steps were performed using an EVG120 automated resist processing system (EV Group). The patterned wafer was post-processed by etching, photoresist stripping, and cutting individual electrodes with a DAD dicing saw (DISCO). The patterned gold electrode purchased from Pine Research was made by screen-printing gold on a ceramic base, with a $2 mm^2$ diameter gold working electrode, a printed Ag/AgCl reference electrode, and a printed gold reference electrode.

### Visualization of the electrodeposited hydrogels

Gold patterned electrodes were submerged in a solution containing HRP (0.1 mg/mL in 0.1 M PB, pH = 6), gelatin (25 mg/mL), and $Ir^{3+}$ (5 mM) for electro-deposition (+1.1 V, 1 min). The resulting hydrogel was then stained with Coomassie Blue solution (0.1% Coomassie® R-250 in 50% ethanol and 10% acetic acid) followed by brief destaining (50% methanol and 10% acetic acid). Brightview images of the hydrogel was taken by the MVX10 upright fluorescence microscope (Olympus).

Overnight BL21 culture was adjusted to OD ~ 12 in 0.85% NaCl solution and stained with SYTO-9 dye (final concentration 8.35 μM) (Thermo Fisher Scientific) for 15 min in the dark. The stained cultures were then mixed 1:1 with 50 mg/mL PEG-SH + 10 mM Fc solution for deposition (+0.7 V, 2 min) on gold-patterned electrodes. Both fluorescence and brightview images were taken by the MVX10 upright fluorescence microscope (Olympus).

### Colorimetric HRP enzymatic activity assay

Gelatin/HRP hydrogel was deposited on two gold patterned electrodes with different surface areas (25 or 100 mm²) as described in the section

above. The resulting hydrogel was then washed in 0.1 M PB (pH = 6) for 10 min before the enzymatic assay. A final concentration of 0.8 mM 2,2′-azino-bis(3-ethylbenzothiazoline-6-sulfonic acid) (ABTS) and 0.003% (0.88 mM) $H_2O_2$ as assay solution mixture was prepared from an ABTS stock solution (16 mM) and 30% $H_2O_2$. The washed hydrogel (on gold patterned electrode) was incubated in 2.2 mL of the assay solution mixture for 4 min at room temperature. The hydrogel was then removed and 1 mL of the assay solution mixture was transferred to a 1-cm cuvette for absorbance reading at 405 nm using an UV-Vis spectrophotometer (Beckman Coulter).

### Fabrication of the optoelectrochemical device

The custom device was designed based on a standard 3-electrode system. The resin-based housing was fabricated with a Mars 3 Pro 3D printer with the standard black resin from ELEGOO (Guangdong, China) and was then attached to a 25 mm × 25 mm ITO-coated glass substrate (Sigma) as the working electrode via photo-curing with the same resin for 3D printing. To cast the agarose salt bridge, the solution containing 0.1% agarose in 1 M KCl was first heated and then pipetted into the central well. After the agarose solidified, 1 M KCl solution was added to submerge the salt bridge. The Ag/AgCl reference electrode (Pine Research) can be inserted to the central well containing the salt bridge via the side opening. A separate custom connector was also fabricated using the 3D printer and resin described above. This connector comprised a platinum wire as the counter electrode and four air nozzles designed to evenly distribute oxygen to the independent wells for hydrogen peroxide production. With the device fully assembled, both the inserted reference electrode and the counter electrode would be immersed in the 1 M KCl solution present in the central salt bridge well. Together with the ITO working electrode, this system formed a complete circuit for electrochemical operation (Fig. 2d).

### Quantification of film thickness

Overnight culture of DH5α-sfGFP was harvested via centrifugation ($3000 \times g$, 15 min) and resuspended in 1× PBS to $OD_{600} = 2 mL^{-1}$. 2× PEG-SH solution was prepared as previously stated and was mixed 1:1 with the *E. coli* solution. 100 μL of the cell/PEG-SH mixture were loaded into the wells of the optoelectrochemical device and electrodeposition (0.8 V) was performed for the indicated duration. After decanting the excess solution and washing thoroughly with PBS, a ZEISS LSM700 confocal microscope was used to obtain Z-stack images of the generated film. Using a 5 μm Z-stack depth, we defined the film thickness based on the distance between the electrode and the boundary layer (i.e., the last Z-stack slice with visible *E. coli*).

### Peroxide generation and quantification

Electrochemical peroxide generation was performed in the optoelectrochemical device with its setup as described above. 150 μL of 20% LB mixed with 80% PBS (henceforth, 20% LB) was added into each well (surface area = 38.5 mm²). The working electrode solution was undisturbed (or, where indicated, stirred via blowing $O_2$ into the wells). The electrodes were connected to a potentiostat (either 700-series CH Instruments or a custom FPGA-based potentiostat). Chronoamperometry, poised at −0.8 V for the indicated duration, was performed to generate hydrogen peroxide. The endpoint charge was recorded for each run.

To quantify the generated peroxide, we used the Pierce® quantitative peroxide assay kit (aqueous) (Thermo Scientific) according to the manufacturer's instructions. Briefly, the working reagent was prepared by mixing one volume of Reagent A with 100 volumes of Reagent B, with at least 200 μL prepared for each sample to be assayed. Ten volumes of the working reagent were added to one volume of sample (typically 200 μL working reagent to 20 μL sample) in a well of a clear-bottomed 96-well plate. The reaction was mixed and incubated for 15–20 min, after which a Spark® microplate reader (Tecan) was

 

used to measure the absorbance at 595 nm. Sample peroxide concentration was calculated by comparison with a standard curve (dilutions of 30% (*w/w*) peroxide) performed on the same day.

## General and coculture electroinduction set-up
Electroinduction experiments were performed in the custom optoelectrochemical device. Deposition of the cell/PEG-SH film was performed as described in previous sections. After thorough washing to remove excess cell/PEG-SH solution, 150 μL of 20% LB was added to each well as the culture media. Peroxide was generated via chronoamperometry as described in sections above, with voltage application for a specified duration (e.g., 1800 s). The optoelectrochemical device was then moved to an incubator (30 or 37 °C, as indicated) for incubation. For all automated experiments, the optoelectrochemical device remained in a custom-made environmental chamber inside the Biospark system with temperature (34 °C) and humidity (≥80%) control. 0.4 ft$^3$ h$^{-1}$ (SCFH) of oxygen was supplied to all samples (including the negative control) during electroinduction for media perturbation and oxygen supply to generate sufficient peroxide.

Samples (i.e., the media immersing the film) were removed at indicated time intervals and sterile filtered for downstream bioassays. For coculture experiments, AI-1 responsive strain (NEB10β + pLasR_S129T-GFPmut3) were grown to mid-log phase and reinoculated to an OD$_{600}$ of 0.025 in 20% LB. Culture was then pipetted into the wells of the optoelectrochemical device in which the 'artificial biofilm' (containing CRISPRa *lasI* cells) was previously deposited. Electroinduction was subsequently performed as described above.

## RT-qPCR
NB101 harboring the plasmids that allow CRISPR activation of GFP (pSC-O108 + pdCas9ω + pMC-GFP) were grown at 37 °C and 250 r.p.m. in LB overnight. The following day, overnight cultures were diluted to OD$_{600}$ = 0.1 and grown at 37 °C and 250 r.p.m until reaching mid-log phase. Peroxide stock solution (10 mM) was then spiked into the cultures to induce the expression of sgRNA. After 2 h of incubation, samples were collected via centrifugation. Total and microRNA were extracted using a miRNeasy Kit (Qiagen) and quantified using a Nanodrop (Thermo Scientific). RT-qPCR was then carried out using primers SW19/SW20 (Supplementary Table 2), the Power SYBR™ Green RNA-to-CT™ 1-Step Kit (Applied Biosystems), and a Quantstudio 7 Flex Real-time PCR system (Applied Biosystems).

## AI-1 quantification
AI-1 quantification was performed by bioluminescence assay. AI-1 reporter cells JLD271 pAL105[72] were grown overnight in LB at 37 °C and 250 r.p.m. shaking with the appropriate antibiotics. The following day, AI-1 solutions (0–84 nM) for the standard calibration curve were prepared in 20% LB. The reporter cells were diluted 500-fold in LB with the appropriate antibiotics. For every experimental replicate, 90 μL of diluted reporter cells and 10 μL of the standard AI-1 solutions were added into the wells of a white-bottom 96-well plate (Corning). Experimental conditioned media samples were prepared similarly after sterile filtering and diluting between two- and thousand-fold to maintain a linear assay range. Microplate with reporter cells and conditioned media samples were incubated at 30 °C and 250 r.p.m. shaking in a Tecan® microplate reader, and its luminescence was measured by the plate reader every 30 min for 3–5 h. The AI-1 concentration of each sample was calculated using the standard curve.

## AI-2 activity assay
Relative AI-2 levels were determined using the *V. harveyi* reporter BB170 bioluminescence assay[73] with slight modifications. AI-2 reporter cells BB170 were grown overnight in AB media at 30 °C and 250 r.p.m.

shaking with appropriate antibiotics. The following day, overnight BB170 culture was diluted 5000-fold in AB media. For every experimental replicate, 180 μL of diluted BB170 culture and 20 μL of conditioned media were added into the wells of a white-bottom 96-well plate (Corning). The microplate was then incubated at 30 °C and 250 r.p.m. shaking in a Tecan® microplate reader, and its luminescence was monitored by the plate reader every 30 min after 3-h incubation. AI-2 activity was calculated by dividing the RLU produced by the reporter after addition of conditioned media by the RLU of the reporter when growth medium alone was added.

## Statistics
The data presentation, sample size of biological replicates (*n*), statistical analysis, and significance of differences are shown in the figures, and the details are described in each figure legend. To determine the statistical significance of differences in the case of multiple comparisons, we used GraphPad Prism 10 (GraphPad Software) or Microsoft Excel (Microsoft) to perform analyses including a two-tailed, unpaired, Student's *t*-test, and one-way or two-way analysis of variance (ANOVA).

## Electrodeposition of HRP/gelatin hydrogel and electrochemical detection of H$_2$O$_2$
We followed the protocol for HRP/gelatin deposition and electrochemical detection of peroxide as described by Li et al. with slight modifications[23]. Deposition of HRP/gelatin was performed on a custom patterned electrode generated through laser-cutting the ITO-coated glass slide into two separate interweaving working electrodes. This zigzag pattern was chosen to ensure the generated peroxide be at the vicinity of the deposited HRP/gelatin hydrogel for detection. A custom 3D-printed housing with one central well exposing the working electrodes and two side openings for the insertion of Ag/AgCl reference electrodes was then attached to the patterned ITO electrode (Fig. 5a, b). A custom connector attached with two Pt wires provided the 3-electrode system with counter electrodes. Upon deposition, a solution containing HRP (1 mg/mL in 0.1 M PB, pH = 6), gelatin (25 mg/mL), and Ir$^{3+}$ (5 mM) was pipetted into the central well of the device followed by applying an oxidative voltage (+1.2 V) for 2 min on working electrode 2 (WE2). After removing the excess HRP/gelatin mixture, a solution of 0.25 mM Fc in 0.1 M PB (pH = 7.4) was added into the well submerging both working electrodes. Electrochemical peroxide generation was performed on working electrode 1 (WE1) by applying −0.8 V for the desired duration as indicated. To detect the peroxide generated from WE1, a constant voltage (0 V vs Ag/AgCl) was imposed on WE2 for 120 s, and the endpoint current output was recorded (Fig. 5d, e).

## Custom-built bioelectronic system (BioSpark) setup
Hardware of the BioSpark system includes a custom two-channel arbitrary waveform potentiostat with simultaneous sampling of up to two PMT sensors, a motorized fiber-coupled fluorescence reader, and a custom environmental chamber. This system is controlled by a custom Windows GUI program.

**Fluorescence measurement and photobleaching module.** The fluorescence module consists of a 470 nm excitation light source and driver (Thorlabs #M470F4 and #DC2200) that sent light to a fiber-coupled filter mount (Thorlabs #FOFMS) via a 1000 μm diameter core multimode fiber (Thorlabs #FT1000EMT-CUSTOM). An OD6 fluorescence filter (Edmund Optics #67-027) was used to set the excitation band. This light was then directed onto the test target via a fiber coupled reflection probe (Thorlabs #RP22). The emission from the target was collected by the other fibers in this bundle and directed to a second fiber coupled filter mount (Thorlabs #FOFMS) which contained two OD6 emission filters (Edmund Optics #67-030). Two filters were used to improve out-of-band blocking with only a

 

modest reduction in the emission signal. The filtered light in turn passed through another fiber (Thorlabs #FT1000EMT-CUSTOM) into a fluorescence enhanced PMT (Thorlabs #PMT2101) where it was converted to an electrical signal that traveled to the analog-to-digital converter (ADC) on the FPGA potentiostat board. A 3-axis motion platform consisting of three stepper motor kits (Thorlabs #KMTS50E) connected with a base plate (Thorlabs #MTS50A-Z8), XY plate (Thorlabs #MTS50B-Z8) and right-angle plate (Thorlabs #MTS50C-Z8) was used to allow x, y, and z-axis movement of the probe. To administer photobleaching, we used the fiber-coupled reflection probe to direct emission light to the sample well for a desired duration (for 15 min in the experiment showcased by Fig. 7). A schematic depicting the setup during fluorescence detection and photobleaching is shown in Supplementary Fig. 17.

**Custom FPGA-potentiostat.** We designed and fabricated a custom FPGA-based two channel potentiostat PCB controller. The core component is a Spartan-7 module (Opal Kelly #XEM7305) and breakout board (Opal Kelly #BRK7305). Power was supplied via an ultralow-noise linear power supply (Acopian #DB5-50). While many components are used on the custom potentiostat board, the three primary components of interest include the (a) femtoampere input bias current op-amps (Analog Devices #ADA4530), (b) the 16-bit multichannel Digital-to-Analog converter (DAC) (Analog Devices #AD5765), and the 18-bit multichannel ADC (Texas Instruments #ADS8698).

**Power source, connection to PC, and custom control software.** General power for the BioSpark system is provided from four power supplies: a 5VDC (CUI # VGS-75C-5), a 12VDC (CUI # VGS-75W-12), a 15VDC (CUI #VGS-75C-15), and a 48VDC (CUI #VGS-75W-48). The system is connected to a PC via USB (Latorice #B07GLP8SD). A custom Windows GUI program written in C# is used to control the system. Functions included adjusting metrics related to the (i) various potentiostat modes (e.g., chronoamperometry mode, cyclic voltammetry mode, manual electroinduction mode, algorithm-controlled electroinduction mode), (ii) potentiostat settings (e.g., voltage, pulse period, slew rate, maximum total charge, etc.), (iii) fluorescence reading (e.g., pulse width, pulse count, measurement interval), and (iv) automated algorithm control (e.g., algorithm/manual control mode, ratio thresholds, expression level target, SMS settings, etc.). Both electrochemical and fluorescence data are saved in a custom binary format but can be exported into an Excel spreadsheet (.xlsx). On the main screen of the program, four real-time diagrams displayed the fluorescence values, potentiostat currents and charges, applied counter voltages, and the CV curve or fluorescence slope ratios. The entire BioSpark system is housed in an opaque enclosure to ensure no stray light enters the test chamber.

**Custom algorithm for 'smart' control of gene expression**
We established a simple model-based algorithm (generic model control)[74] to monitor gene expression for the surface-assembled recombinant *E. coli* producing either transcription or translation products as represented by the emitted fluorescence (*F*). The process was described by a dynamic model:

$$F = g(\mathbf{x}, \mathbf{u}, t) \tag{1}$$

where $\mathbf{x}$ is the state vector, $\mathbf{u}$ is a vector of potentiostat inputs, and $t$ is time. Process input $u(t)$ can be described by Eq. 2:

$$u = h(\mathbf{a}, t) \tag{2}$$

where $\mathbf{a}$ is the current resulted from electroinduction, and $t$ is time. An algorithm was then developed from this model to track the output fluorescence and control the system (Supplementary Fig. 12):

Since the fluorescence were taken at a fixed time interval (15 min/ 0.25 h), we defined the slope ($S$; RFU $\times 4 \times h^{-1}$) as the difference between two neighboring fluorescence measurements as described by Eq. 3.

$$S_n = F_{(n+1)} - F_n \tag{3}$$

The algorithm stores and updates the value of the maximum slope ($S_{max}$) to serve as a reference point for the progression of the fluorescence level. The change in fluorescence levels was determined by the ratio ($R$) between the current slope and $S_{max}$, as described by Eq. 4.

$$R_{(n-1)} = S_n \div S_{max} \tag{4}$$

If two consecutive ratios fall below the user-defined ratio value (set as 0.4 in both Fig. 6, Fig. 7, and Supplementary Fig. 13), the algorithm considers the ratio threshold is met and will initiate electroinduction by sending a command to the potentiostat. $S_{max}$ will return to 0 and a new cycle will start subsequently. Variables such as the ratio threshold, duration of voltage application, or the total charge applied can be set in the GUI program.

**Custom-built, bio-electrochemical platform ('actuation checkpoint') setup**
An individual custom two-channel arbitrary waveform potentiostat was built through FPGA board fabrication and assembly, like the BioSpark potentiostat described in previous sections. Similarly, the system is connected to a PC via USB, and a custom Windows GUI program written in C# is used to control the system. Potentiostat settings for both channels (WE1 and WE2) such as voltage, pulse period and control settings such as current threshold can be adjusted in the program. Electrochemical data are saved in a custom binary format but can be exported into an Excel spreadsheet (.xlsx).

**Custom algorithm for 'smart' control of enzymatic activity**
A simple algorithm was built to control the 'actuation checkpoint' bio-electrochemical platform; the system diagram for the connecting both gene expression (by the local BioSpark) and enzymatic activity (by the remote enzymatic 'actuation checkpoint') is depicted in Supplementary Fig. 15. After receiving the initiation message from the local BioSpark system, the algorithm then commands the 'actuation checkpoint' (situated at a remote location) to run a pre-set program: apply −0.8 V on WE1 for 600 s, followed by 0 V on WE2 for 120 s. It then compares the value of the output current to that of the user-defined current threshold: if the output current does not exceed the threshold, a message is sent to the local BioSpark system to administer electroinduction immediately after the upcoming fluorescence measurement (for a user-defined duration or charge); otherwise, the algorithm sends a SMS verification message to alert the human users and seek permission to terminate the experiment. If 'Y' is received from the human users, a termination cue will be sent to the local BioSpark system to initiate photobleaching immediately after the upcoming fluorescence measurement. If 'N' is received form the users, the algorithm then prompts the 'actuation checkpoint' to run the pre-set program once more.

**Reporting summary**
Further information on research design is available in the Nature Portfolio Reporting Summary linked to this article.

## Data availability
Sequences of relevant genetic parts from all plasmids generated in this study can be found in the Supplementary Information and Supplementary Data 1. All other data and information are available upon

request from the corresponding author. Source data are provided with this paper.

## Code availability

The complete code and packaged software are available from the corresponding author upon request.

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

## Acknowledgements
The authors would like to acknowledge partial support of this work by the National Science Foundation (MBC#2227598, CBET#1932963), the Department of Energy (BER# SCW1710), the Defense Threat Reduction Agency (HDTRA1-19-1-0021), and the Gordon and Betty Moore Foundation (#11395). As well as the National Institutes of Health (T32 Award, Training Program in Host-Pathogen Interactions, 5T32AI089621-07, to S.W.) and the A. James and Alice B. Clark Foundation (Clark Doctoral Fellowship, to S.W.).

## Author contributions
S.W., C.-Y.C., C.-Y.T., E.V., and W.E.B. conceived and designed the experiments. S.W., C.-Y.C., C.-Y.T., J.L., E.K., and F.R.Z. performed the experiments. C.-Y.C. designed and fabricated custom devices. J.R.R., S.W., and C.-Y.C. designed and built the BioSpark system. S.W. and J.R.R. developed the custom algorithm, and J.R.R. wrote the code for the BioSpark system. S.W., C.-Y.T., and W.E.B. analyzed and interpreted the data. S.W., G.F.P., and W.E.B. wrote the manuscript with contributions from all authors.

## Competing interests
The authors declare no competing interests.
