## [Peer Review File · Nature Communications]

REVIEWER COMMENTS

Reviewer #1 (Remarks to the Author):

Summary

In this manuscript, Wang et al. have described and characterized an electrogenetic device allowing cell-to-cell communication at different hierarchical levels from proteins to cell consortia by programming a redox-mediated microelectronic system. They biofabricated horseradish peroxidase (HRP) and engineered *E. coli* cells on electrodes to control their activities on chip electronically. Bacterial electrogenetic tools, *oxyRS* stress regulons were used to demonstrate how cells can be electrostimulated and recorded for feedback control of reporter expression. Importantly, they realized algorithm-facilitated automated control in protein-based and cell-based subsystems by designing customized electrochemical devices for study, communication and behaviour control of biological systems.

This group has developed several redox-based systems since years ago. The system described in this manuscript pushed the field of electrogenetics forward. Using electronic devices to control gene expression is not new, while it's quite fantastic to set up a closed-loop circuit using redox mediators that bridges the electronic signals and biomolecules for feedback control and communications between physical devices and biological systems. This holds the potential to be further developed to connect biological systems and human wearable interfaces in the future age of the "Internet of Life".

Some specific comments are listed below:

1. The abstract and introduction part should be more structured. Introducing some background in the abstract may help readers to better understand the rationale of this work. Recent relevant advances in the electrogenetics field should be included, like Huang et al., *Nat. Metab.* 2023. Additionally, figure 1 looks a simplistic, it's more like a graphic abstract.
2. In all figures, no statistical analysis is presented. To support and strengthen your conclusions, scientific statistical significance should be calculated.
3. In Fig. S4 (i), two times of peroxide induction resulted in higher AI-1 levels, while three and four times of induction decreased the levels compared to two times of induction, although they are still higher than one induction. What would be the explanation for that? could it be the cytotoxicity of accumulated peroxide?
4. In Fig. S4 (ii), it seems that there is no significant difference between the uninduced and one-time induction group. The claim in line 192 is not so convincing. More data points and tests are needed.
5. In figure 3, only the percentage of electroinduced cells was counted. How about the intensity of expressed GFP? In Fig. S2, the authors showed the fluorescence intensity is kind of proportional to the

concentration of H₂O₂. While in fig 3c,d, only a fraction of fluorescing cells were presented. Moreover, the control is not scientific. 200 μM of H₂O₂ shows a similar fraction as 10 min of induction duration, but it seems doesn't mean that cells in 10 min of induction duration generate 200 μM of H₂O₂. As we can imagine, the higher concentration of H₂O₂, the lower the cell viability. Probably cells treated with 500 μM of H₂O₂ show a similar fraction as 5 min duration. So this control could be not appropriate. Fraction is important, but maybe intensity is also needed.

Minor:

1. In line 409, it seems there is a missing word in "reduced oxygen (H₂O₂)".
2. In supplementary Fig. 4, fold changes and statistical significance should be presented in figures, as declared in text in lines 173-177, otherwise readers would be confused about which one it meant.
3. There is a missing reference for supplementary Fig. 6.
4. It's a bit strange that supplementary fig. 10-13 appear earlier than the other supplementary figures. For example, supplementary fig. 10 comes after supplementary fig. 1 and before any of the other supplementary figures. So as well fig. S11, S12, S13.

Reviewer #2 (Remarks to the Author):

In this manuscript, the authors have constructed genetic circuits to convert electronic information into biological responses. This includes the development of an electronic "language translator" capable of modulating cellular populations. The authors utilized electro-genetics in conjunction with the oxyRS stress regulon of *Escherichia coli* to illustrate how both engineered and native cells can be electronically manipulated and monitored. This research holds significant relevance for the synthetic biology community. To enhance the manuscript, the authors should consider addressing the following comments:

Major Comments:

1. The authors have previously published a study* in which they employed redox-responsive promoters to achieve user-specified control over biological function. In this prior work, they developed a system driven by eCRISPR for the spatiotemporal transmission of redox commands, which were subsequently decoded by gelatin-encapsulated *E. coli*. Their findings demonstrated the potential of redox communication for advancing biohybrid microelectronics and enhancing control over biological

functions. It appears that the current study may be considered an extension of their previous work. Therefore, the authors should provide a clear justification for the novelty of the research presented in the current manuscript.

*(Bhokisham, N., VanArsdale, E., Stephens, K.T., Hauk, P., Payne, G.F. and Bentley, W.E., 2020. A redox-based electrogenetic CRISPR system to connect with and control biological information networks. Nature communications, 11(1).

2. Figure 1: The drawing implies the use of electromagnetic waves for manipulating cells which is not the case in this study. Please revise the figure to avoid confusion.

3. The authors have employed oxidative stress-responsive promoters, such as oxyRS, in the construction of a CRISPR-based electro-genetic circuit. Notably, these oxidative stress promoters typically exhibit basal or leaky expression levels. The manuscript would benefit from an elucidation of how the authors achieved precise regulation within their electro-genetic circuit, particularly in response to specific inducers. This discussion can be added to the manuscript.

4. In this study, the authors developed an artificial biofilm consisting of E. coli entrapped in thiolated-polyethylene glycol (PEG-SH). The similarities and differences between this artificial biofilm and naturally occurring biofilms should be elucidated in the manuscript.

5. In Line 128, the authors stated that “we applied reducing potential (-0.8 V) to electroinduce CRISPRa in the cells electrically assembled on the ITO electrode surface (Fig. 3b). Using confocal microscopy, we found that ~70 % of electro induced cells expressed GFP, while ~35% of electro assembled cells expressed GFP when 200 μ M peroxide was introduced to the growth media above the film, and < 5% of the cells expressed GFP in negative controls. Interestingly, the fraction of fluorescing cells increased with charge”. A discussion on the cellular regulatory mechanism that increases the GFP intensity with increasing voltage and charge (electrical induction) is required. Additionally, it is not clear what happens if we increase the reducing potential above 0.8V. The authors should discuss this in the manuscript.

6. Figure 3c, d: It is not clear why the population percentage is used as a quantification metric instead of single-cell fluorescence data. Any potential solutions to produce a homogeneous population of electroinduced cells should also be provided to improve the reproducibility of the platform.

7. Figure 5 and 6: The authors should provide reasoning for not using the BioSpark system to control quorum sensing mechanisms in the cells. This would be a more powerful demonstration of the system, further strengthening the claims in the manuscript. For instance, the authors can clone GFP downstream

of QS modules and calibrate the GFP signals according to the activity levels of QS modules, then use them for feedback-control by Biospark.

8. The authors should provide diagrammatic illustrations of all constructed genetic circuits with genetic components as supplementary figures. These visual aids will enhance the reader's comprehension of the manuscript.

9. In the discussion section, authors should present the limitations of current research and discuss potential areas for future work.

Minor comments:

1. Line 57: The extra spacing should be removed - Fig. 1b.

2. Authors should maintain consistency in their usage of "Fig." or "Figure" throughout the entire manuscript. For instance, in lines 73, 75, 83, 280, 285, and 288.

3. The authors should rephrase the Line 86 for clarity: "4-6 h after 7 E. coli BL21 that constitutively secrete quorum sensing (QS) autoinducer AI-2 were assembled onto electrodes with different surface areas (25 mm² vs 100 mm²), the growth media submerging the larger electrode contained ~6 times the AI-2 activity of the smaller electrode, suggesting preservation and proportionality of cell function with electrode area."

4. Figure S2: What is the KD for these 2 response curves? Single-cell data would be more informative.

5. Line 203: The bioluminescence data as mentioned in this line and Figure 4 is missing.

6. Line 245: Please expand FPGA.

7. Line 246: Please define GUI.

8. Line 498: Please correct the typo "inducation".

Reviewer #1 (Remarks to the Author):

Summary

In this manuscript, Wang *et al.* have described and characterized an electrogenetic device allowing cell-to-cell communication at different hierarchical levels from proteins to cell consortia by programming a redox-mediated microelectronic system. They biofabricated horseradish peroxidase (HRP) and engineered *E. coli* cells on electrodes to control their activities on chip electronically. Bacterial electrogenetic tools, *oxyRS* stress regulons were used to demonstrate how cells can be electrostimulated and recorded for feedback control of reporter expression. Importantly, they realized algorithm-facilitated automated control in protein-based and cell-based subsystems by designing customized electrochemical devices for study, communication and behaviour control of biological systems.

This group has developed several redox-based systems since years ago. The system described in this manuscript pushed the field of electrogenetics forward. Using electronic devices to control gene expression is not new, while it's quite fantastic to set up a closed-loop circuit using redox mediators that bridges the electronic signals and biomolecules for feedback control and communications between physical devices and biological systems. This holds the potential to be further developed to connect biological systems and human wearable interfaces in the future age of the "Internet of Life".

We thank and appreciate the reviewer for the insightful comments and suggestions. Our responses to the specific comments can be found below.

Some specific comments are listed below:

1. The abstract and introduction part should be more structured. Introducing some background in the abstract may help readers to better understand the rationale of this work. Recent relevant advances in the electrogenetics field should be included, like Huang *et al.*, *Nat. Metab.* 2023. Additionally, figure 1 looks a simplistic, it's more like a graphic abstract.

We have restructured our abstract and introduction to include more recent and relevant works in the field of electrogenetics (please see the texts marked in red). Additionally, the abstract was reworded to provide more of the motivation as well as experimental detail. Further, we revised the title and edited **Fig. 1**. We believe the revised versions greatly clarify our motivation and its perception.

2. In all figures, no statistical analysis is presented. To support and strengthen your conclusions, scientific statistical significance should be calculated.

We have provided statistical analysis as requested (see for example, revised **Fig. 3 and 4**). Statistical methods used in the study are provided in the **Methods** section.

3. In Fig. S4 (i), two times of peroxide induction resulted in higher AI-1 levels, while three and four times of induction decreased the levels compared to two times of induction, although they are still higher than one induction. What would be the explanation for that? Could it be the cytotoxicity of accumulated peroxide?

We thank the reviewer for this insightful question. There are several factors at play that we feel can influence the levels of the autoinducers, particularly in later stages of cell culture: (i) as noted by the reviewer, there could be some level of cytotoxicity from the additional hydrogen peroxide; (ii) there could be a reduction in the synthesis of autoinducers as the cells enter the end of log growth; and (iii) there could be some uptake by the cells in the late stages, noting that the later OD's are much higher than in the cases of 3 & 4 h additions. While we cannot be certain, we do not attribute attenuated autoinducer production to peroxide toxicity (see next response).

To help to explain our conclusions, we have provided additional OD₆₀₀ data (**Supplementary Fig. 7d (iii)**) to the original Supplementary Fig. 4 (now **Supplementary Fig. 7**). In this figure, we can see tapering of the cell growth as well as the relative decrease in growth for the cultures with more peroxide additions later in the cultures. Specifically, at hour 7, the uninduced samples (OD₆₀₀ ~ 2.32) had a slightly higher OD₆₀₀ than the samples with three (OD₆₀₀ ~ 2.16) or four (OD₆₀₀ ~ 2.12) inductions. Thus, both the added peroxide and the overall decrease in metabolic activity (which would impact LasI-mediated AI-1 synthesis) in the late stages of the culture could be contributing factors. Finally, there could also be some uptake in AI-1. We have added text to the **Supplemental Information** to convey these points.

4. In Fig. S4 (ii), it seems that there is no significant difference between the uninduced and one-time induction group. The claim in line 192 is not so convincing. More data points and tests are needed.

We have run several sets of experiments to obtain the QS (AI-1 and AI-2) data that are depicted in this manuscript (now **Supplementary Fig. 7d**). It is important to point out that the assays for these signal molecules are biological (reported by sensor cells) and have

inherent variability. Researchers typically “fold induction” as opposed to the activity. In the figure below, we show AI-2 data for a completely separate experiment, among many, that were run to make sure that the trends depicted in our manuscript are reproducible. The bars indicated represent the means of biological replicates of the same samples.

To compare datasets between campaigns, we have normalized our data to the uninduced controls, and in the plot below (**Fig. 1c**), combined this with the data from **Supplementary Fig. 4** of our original manuscript. We have provided statistical relevance here, and validation of the observed trends.

Figure 1. (a) (From the original *Supplementary Fig. 4*): AI-2 activity secreted from ‘bilingual’ cells 7 h post-reinoculation in liquid culture. **(b)** (From another individual experiment): AI-2 activity secreted from ‘bilingual’ cells 7 h post-reinoculation in liquid culture. **(c)** Normalized AI-2 activity from (a) and (b). Comparisons in (a) and (b) used ordinary one-way ANOVA. Comparisons in (c) used unpaired, two-sided *t* test. * $P < 0.05$, ** $P < 0.01$, *** $P < 0.001$, **** $P < 0.0001$

Because our experimental campaigns involve many measurements that are themselves subject to error, we believe it is most appropriate to include the comprehensive QS AI-2 data set from one of the replicates in an unnormalized manner, rather than a normalized amalgamation of many experiments consisting of many measures. In this way, should someone want to replicate the work, the actual data are provided rather than a normalized value.

5. In figure 3, only the percentage of electroinduced cells was counted. How about the intensity of expressed GFP? In Fig. S2, the authors showed the fluorescence intensity is

kind of proportional to the concentration of H₂O₂. While in fig 3c,d, only a fraction of fluorescing cells were presented. Moreover, the control is not scientific. 200 μM of H₂O₂ shows a similar fraction as 10 min of induction duration, but it seems doesn't mean that cells in 10 min of induction duration generate 200 μM of H₂O₂. As we can imagine, the higher concentration of H₂O₂, the lower the cell viability. Probably cells treated with 500 μM of H₂O₂ show a similar fraction as 5 min duration. So this control could be not appropriate. Fraction is important, but maybe intensity is also needed.

The reviewer raises on a very important issue that also allows us to reinforce the novelty (and value) of the methods we have developed. In short, there is no true positive control for the data in this plot, because the addition of 200 μM H₂O₂ to the liquid above the gel does not mean the cells also experienced 200 μM. We have addressed this in some detail below and have altered the manuscript to make this clear (please see the texts marked in red at **line 135** and **472**).

First, we wish to demonstrate that it is indeed peroxide that induced the cellular fluorescence response by adding 200 μM peroxide into the submerging media to activate the *oxyS* promoter. This concentration was chosen because we have shown in **Supplementary Fig. 4a** that it can elicit a saturated fluorescence response to the identical cells in liquid suspension culture. However, as noted by the reviewer, it appears that 200 μM peroxide results in a similar biological response as 10 min of electroinduction. When one looks at **Fig. 2e**, one will find that roughly 25 μM peroxide is generated after 10 min of applied voltage to an equivalent sample volume in our device. Our data shows that the same biological result can occur with what appeared to be ~8-fold *less* peroxide. Moreover, the actual peroxide generated in the presence of cells in a gel can be a lot less than 25 μM (as expected from **Fig. 2f**). We believe it is inappropriate to directly equate minutes of induction of cells in a hydrogel with the amount of peroxide generated from the same electrode otherwise suspended in buffer or media. Importantly, the actual peroxide-generating current is *measured* during the induction. This reflects the rate of generation of peroxide (see **Fig. 2f** and **Supplementary Fig. 15, 17**).

Integrating current over time provides an indicator of the amount of peroxide generated at the electrode. It will not reflect the amount of peroxide in solution as cells (and the gel) will consume this peroxide (see the profiles in **Fig. 2f**, for the relative contributions of the cells and the gel for attenuating peroxide). That is, the current in **Fig. 2f** in the case of no gel (shown in teal) starts high (~15 μA), drops rapidly in the first minute and steadies out

at 5 min. The current never drops below 5 μA . The area under the curve (charge) reflects the amount of peroxide produced. With the addition of the gel (orange), there is less current and less area under the curve. The attenuated area indicates that less peroxide was generated. Finally, when one examines the current in the presence of gels containing metabolizing cells, there is very little current indicating depleted levels of oxygen at the electrode. Only when oxygen was added into the system did the current increase from zero, indicating that peroxide was produced starting at around 120 seconds, and at levels less than without cells. The takeaway from this analysis, albeit non quantitative, is that both the cells and the gel decrease the availability of dissolved oxygen at the electrode and thus limit the amount of peroxide generated upon the application of voltage (when compared to a solution with an electrode). So, in the case of 10 minutes of applied voltage to solutions, there was $\sim 25 \mu\text{M}$ peroxide and we expect the cells in the gel see far less peroxide than $200 \mu\text{M}$, yet they exhibited similar induction levels! That said, for the identical reasons, we feel that the $200 \mu\text{M}$ addition case is also not a true positive control, as noted by the reviewer.

In sum, as the reviewer has alluded to, perhaps because the biological result between the $200 \mu\text{M}$ bolus addition case and the 10-minute electroinduction case resulted in similar levels of GFP, these cells “saw” about the same level of peroxide on average. We cannot measure this independently, however.

Both reviewers expressed concern relative to our using the fraction of cells that are induced as opposed to total fluorescence. We agree this is atypical and have provided rationalization below. While not stated, this stems from the development of our optical system and our desire to measure fluorescence within the layers of the gel, irrespective of the fluorescence and volume of liquid above the gel. We are confident in our measurements and have provided additional evidence that measurements in the gel can be summed to obtain total fluorescence.

That is, in our previous work¹ we have found that the fluorescence trajectory of cells is determined by the concentration of the inducer when first exposed. We reexamined all our data to quantify the per “cell” fluorescence, the size of the pixels representing “cell” or clusters of cells, as well as the sum of the fluorescence in a field of view. We show this in **Supplementary Fig. 6**. Additional data showing the actual number of ON cells and total cells ($\text{ON}\% = \text{ON cells}/\text{total cells}$) can be found in **Supplementary Fig. 6** as well. Here, we show that the mean “per cell” fluorescence (**Supplementary Fig. 6c**) follows a similar

trend to the number of ON cells (**Supplementary Fig. 6e**), but the differences are nowhere near as significant (aligning with previous indications that the fluorescence per cell is fairly constant once it is turned “ON”). We also found that the pixel size was statistically uniform (**Supplementary Fig. 6d**). We were interested in this because it is not formally correct that the dots in the images are individual cells. We have no specific measure of the numbers of cells in a dot. Hence, the pixel size represents the quantify of cells in a spot. Thus, when we sum the total fluorescence associated with the spots, the result (**Supplementary Fig. 6b**) tracks well with the fraction of cells induced.

Additionally, to address the reviewer’s concern of the cytotoxicity of peroxide, we provided data depicting OD₆₀₀ or total cell count of eCRISPRa cells cultured in suspension as well as in film (see **Supplementary Fig. 4a** and **6f**, respectively). We found samples receiving higher concentrations of peroxide or longer electroinductions show only slightly lower OD₆₀₀ or cell count, suggesting peroxide exhibited minimal toxicity to the eCRISPRa cells. We have added this discussion to the **Supplementary Information**.

Minor:

1. In line 409, it seems there is a missing word in “reduced oxygen (H₂O₂)”.

We changed “reduced oxygen (H₂O₂)” to “reduced oxygen (O₂ is reduced to H₂O₂)” (changes are marked in red) so as to avoid confusion.

2. In supplementary Fig. 4, fold changes and statistical significance should be presented in figures, as declared in text in lines 173-177, otherwise readers would be confused about which one it meant.

We have now included additional information including fold changes in the original Supplementary Fig. 4 (now **Supplementary Fig. 7**).

3. There is a missing reference for supplementary Fig. 6.

We have added the figure callout for the original Supplementary Fig. 6 (now **Supplementary Fig. 11**) at **line 252** (please see the texts marked in red).

4. It’s a bit strange that supplementary fig. 10-13 appear earlier than the other supplementary figures. For example, supplementary fig. 10 comes after supplementary fig.

1 and before any of the other supplementary figures. So as well fig. S11, S12, S13.

We have rearranged our **Supplementary Figures** section so that all supplementary figures appear in order in the main text.

Reviewer #2 (Remarks to the Author):

In this manuscript, the authors have constructed genetic circuits to convert electronic information into biological responses. This includes the development of an electronic “language translator” capable of modulating cellular populations. The authors utilized electro-genetics in conjunction with the *oxyRS* stress regulon of *Escherichia coli* to illustrate how both engineered and native cells can be electronically manipulated and monitored. This research holds significant relevance for the synthetic biology community. To enhance the manuscript, the authors should consider addressing the following comments:

We thank and appreciate the reviewer for the insightful comments and suggestions. Our responses to the specific comments can be found below.

Major Comments:

1. The authors have previously published a study* in which they employed redox-responsive promoters to achieve user-specified control over biological function. In this prior work, they developed a system driven by eCRISPR for the spatiotemporal transmission of redox commands, which were subsequently decoded by gelatin-encapsulated *E. coli*. Their findings demonstrated the potential of redox communication for advancing biohybrid microelectronics and enhancing control over biological functions. It appears that the current study may be considered an extension of their previous work. Therefore, the authors should provide a clear justification for the novelty of the research presented in the current manuscript.

*(Bhokisham, N., VanArsdale, E., Stephens, K.T., Hauk, P., Payne, G.F. and Bentley, W.E., 2020. A redox-based electrogenetic CRISPR system to connect with and control biological information networks. *Nature communications*, 11(1).

We appreciate the reviewer’s concern and wish to highlight the novelty of our work. This is the first report of electronic sensing and control of biology at the molecular, cellular, and population levels. We have modified our introduction and results sections to make these contributions more clear.

We provide more detail as follows:

First, we developed an electrodeposition system that enables electronic induction of genetic circuits, as well as real-time feedback control, without mediators and within gels.

Second, we show this for both engineered cells and enzymes. An important aspect is that the same electrodes for assembly provide real time data for surveillance and control (i.e., the H₂O₂ generation and HRP activity are measured via current, and the metabolic activity of the entrapped cells is also indicated via current).

Third, we further illustrate benefits of electronic surveillance and control by connecting these systems and include user (through SMS text messaging) oversight.

Fourth, along the way, we show how we created a “bilingual” cell using simultaneous CRISPRa and CRISPRi. We suggest that the novelty and significance of the paper is much greater than the advances of electronically induced multiplexed CRISPR activity in gels.

We believe our results significantly advance the fields of synthetic biology and bioelectronics, and for the first time, demonstrate an electronically-inducible, closed-loop feedback control system that is heretofore not shown.

2. Figure 1: The drawing implies the use of electromagnetic waves for manipulating cells which is not the case in this study. Please revise the figure to avoid confusion.

The first reviewer had similar concerns. We have revised the figure accordingly.

3. The authors have employed oxidative stress-responsive promoters, such as *oxyRS*, in the construction of a CRISPR-based electro-genetic circuit. Notably, these oxidative stress promoters typically exhibit basal or leaky expression levels. The manuscript would benefit from an elucidation of how the authors achieved precise regulation within their electro-genetic circuit, particularly in response to specific inducers. This discussion can be added to the manuscript.

We thank the reviewer for this concern. We agree that oxidative stress promoters exhibit basal or leaky expression levels, which is readily observed in our data. To reduce leakiness with the *oxyR* and the *oxyS* promoters, we engineered the RBS site downstream of the *oxyS* promoter to ensure a tighter translation (e.g., we used RBS33 from the iGEM catalog in plasmids pOxy-sfGFP and pOxy-sfGFP-AAV). Since gRNA does not require the subsequent translation, we also used other methods to reduce leakiness. For example, (i) the gRNA plasmid (pSC-O108) is a low-copy number plasmid; (ii) we added a degradation tag after *lasI* in pMC-lasI-LAA to lower overall AI-1 production. We also showed, albeit

previously, that one can use CRISPRi to downregulate otherwise upregulated defense response genes². Finally, we note that one benefit of an electrogenetic system is that inputs and outputs are electronic and easily processed so that variances can be calculated, refined, and thresholded as needed.

4. In this study, the authors developed an artificial biofilm consisting of *E. coli* entrapped in thiolated-polyethylene glycol (PEG-SH). The similarities and differences between this artificial biofilm and naturally occurring biofilms should be elucidated in the manuscript.

We have included a short descriptor of “artificial biofilms” in the revised manuscript starting at **line 348** (please see the texts marked in red) and its distinction from biofilms found in natural settings. To elaborate, naturally-occurring biofilms employ extracellular polymeric substances (EPSs) which are a conglomeration of proteins, lipid, and DNAs to embed the cells. In an analogous fashion, the PEG-SH polymer builds a matrix to entrap cells, but the thiol linkages resemble more the extracellular matrices found in mammalian systems (e.g., the mucus layer^{3,4}). Bacteria in naturally-occurring biofilms also elicit shifts in their phenotype to live as a community. Although we provided some data regarding the physiological nature of the in-film bacteria (e.g., their growth and AI-2 profile are depicted in **Supplementary Fig. 7** and **12**, respectively), more studies (especially transcriptome studies) can help characterizing the physiology of bacteria entrapped in these “artificial biofilms”, as noted in our additional text.

5. In Line 128, the authors stated that “we applied reducing potential (-0.8 V) to electroinduce CRISPRa in the cells electrically assembled on the ITO electrode surface (Fig. 3b). Using confocal microscopy, we found that ~70 % of electro induced cells expressed GFP, while ~35% of electro assembled cells expressed GFP when 200 μ M peroxide was introduced to the growth media above the film, and < 5% of the cells expressed GFP in negative controls. Interestingly, the fraction of fluorescing cells increased with charge”. A discussion on the cellular regulatory mechanism that increases the GFP intensity with increasing voltage and charge (electrical induction) is required. Additionally, it is not clear what happens if we increase the reducing potential above 0.8V. The authors should discuss this in the manuscript.

The increase in GFP intensity is a direct result of increased hydrogen peroxide. The genetic regulation is mediated by electroinduction of the *oxyRS*-regulated gRNA transcription. A more detailed description on eCRISPRa can be found in **Fig. 3**.

In turn, peroxide is generated through the partial reduction of dissolved oxygen, and in theory any voltage that exceeds the redox potential of peroxide (+0.085 V vs. Ag/AgCl, pH = 7) can lead to the generation of peroxide⁵. Because of its high activation overpotential, peroxide production usually requires a negative polarization potential (like the -0.8 V chosen by us). We included additional data and descriptions in **Supplementary Fig. 2** (please note that we have rearranged the order of our supplementary figures) to show the effect of voltage during electroinduction. A maximum level of peroxide was detected by applying -0.8 V. We also showed in **Fig. 2e** that peroxide level increases with charge (the product of voltage and time). Typically, with the same amount of time, peroxide production will increase indefinitely with increasing charge. As seen in **Supplementary Fig. 2**, higher voltages (-0.9 V and -1 V), however, did not produce more peroxide. We think this can be attributed to the reduction of indium in the indium tin oxide electrode material that is indicated by a color change of the ITO electrode after the application of a high reducing potential⁶. We have added this discussion to the **Supplementary Information**.

6. Figure 3c, d: It is not clear why the population percentage is used as a quantification metric instead of single-cell fluorescence data. Any potential solutions to produce a homogeneous population of electroinduced cells should also be provided to improve the reproducibility of the platform.

The reviewer, like the previous reviewer, has asked an insightful question. We recapitulate the previous response here. Induction of genes in cells is a heterogeneous process, a result of different concentrations of inducers interacting with different cells, changes in the signal transduction within cells, variability in the transcriptional and translational responses to the inducers, etc. As noted above, we have added new quantification of the fluorescence of the individual cells in our experiments by examining the mean gray values in our confocal images (**Fig. 3c**) and found their fluorescence intensities (**Supplementary Fig. 6c**) varied somewhat similarly to the number of ON cells (**Supplementary Fig. 6e**), but not as statistically significantly. Correspondingly, we have provided another plot (**Supplementary Fig. 6b**) of the total fluorescence contained within the images in **Fig. 3c** and show how this shares a similar increasing trend as the fraction of “ON” cells. As noted above, additional explanatory text is included in **Supplementary Fig. 6**.

While this addresses the reviewer’s concern, we had previously considered other possible solutions to this very issue. For example, we considered retrieving the cells from the PEG-

SH gels, but this would require strong reductants like DTT to reduce the disulfide bonds; we think that this treatment will also affect the cell's viability and fluorescence measurement.

7. Figure 5 and 6: The authors should provide reasoning for not using the BioSpark system to control quorum sensing mechanisms in the cells. This would be a more powerful demonstration of the system, further strengthening the claims in the manuscript. For instance, the authors can clone GFP downstream of QS modules and calibrate the GFP signals according to the activity levels of QS modules, then use them for feedback-control by Biospark.

We appreciate this reviewer's comment and agree that we could use the CRISPR modules in the BioSpark. In essence, this is what we have done, but it is not entirely clear. We have modified accordingly.

To be specific, in **Fig. 3f**, we demonstrated that the AI-1 generated from the in-film, electroinduced eCRISPR cells is recognized by the planktonic AI-1 reporter cells, resulting in a higher fluorescence output. This experiment was conducted using a custom 4-well optoelectrochemical device that is both cultured and monitored in the Tecan Spark® microplate reader. In essence, this optoelectrochemical device is the inner section of the BioSpark (see the image below, **Fig. 2**), but it can be removed from the BioSpark (that is “tuned” to record GFP in the gels specifically) and inserted into a spectrophotometer (that reads bulk fluorescence). The same electronic processes can be used (the insert is connected to a potentiostat and can be induced via application of reducing potential). Data in the original Supplementary Fig. 6 (now **Supplementary Fig. 10**) and newly added **Supplementary Fig. 11** jointly show the BioSpark system can perform on par with a commercial device (i.e., Tecan Spark®) under our experimental conditions to read the fluorescence emitted from both the film and the supernatant. Our motivation for the use of the insert, maintained outside of the BioSpark was that the QS were executed so that the signaling was perceived in the fluids above the gel. Here the total fluorescence is better measured by the Tecan as the BioSpark measurements are tailored (by placement and calibration of the fluorescence probe) to resolve fluorescence within the gels. Our comparative data in **Supplementary Fig. 11** supports the equivalence.

Figure 2. Schematic depicting the inner section of BioSpark where the optoelectrochemical device is placed.

8. The authors should provide diagrammatic illustrations of all constructed genetic circuits with genetic components as supplementary figures. These visual aids will enhance the reader's comprehension of the manuscript.

We have included plasmid maps for all the plasmids generated in this study in **Supplementary Fig. 18**.

9. In the discussion section, authors should present the limitations of current research and discuss potential areas for future work.

We are excited by this comment and have included more discussion regarding the limitations of our research and potential areas for future work in the **Discussion** section (please see the texts marked in red).

Minor comments:

1. Line 57: The extra spacing should be removed - Fig. 1b.

The extra spacing is now removed.

2. Authors should maintain consistency in their usage of "Fig." or "Figure" throughout the entire manuscript. For instance, in lines 73, 75, 83, 280, 285, and 288.

We apologize for the inconsistency. All figure callouts are now reformatted to “Fig. xxx” instead of “Figure xxx”.

3. The authors should rephrase the Line 86 for clarity: “4-6 h after 7 E. coli BL21 that

constitutively secrete quorum sensing (QS) autoinducer AI-2 were assembled onto electrodes with different surface areas (25 mm² vs 100 mm²), the growth media submerging the larger electrode contained ~6 times the AI-2 activity of the smaller electrode, suggesting preservation and proportionality of cell function with electrode area.”

We apologize for the confusion. The original sentence has been rephrased as “4-6 h after *E. coli* BL21 (they constitutively secrete quorum sensing (QS) autoinducer AI-2) were assembled onto electrodes with different surface areas (25 mm² vs 100 mm²), the growth media submerging the larger electrode contained ~6 times the AI-2 activity of the smaller electrode (**Fig. 2b (ii)**). This result suggests preservation and proportionality of cell function with electrode area.”

4. Figure S2: What is the KD for these 2 response curves? Single-cell data would be more informative.

We have reformatted the original Supplementary Fig. 2a (now **Supplementary Fig. 4a**) and included the half maximal effective concentration (EC₅₀) of hydrogen peroxide (H₂O₂) for inducing CRISPRa response in liquid cultures. Our fluorescence data represents the overall fluorescence divided by the cell concentration (OD₆₀₀) of a homogenous culture.

5. Line 203: The bioluminescence data as mentioned in this line and Figure 4 is missing.

We apologize for the confusion. Both measurements to determine the level of AI-1 and AI-2 secreted by the “bilingual” cells employed luminescent reporter cells. We have revised the manuscript to make this more clear (please see the texts marked in red at **line 206** and **495**).

6. Line 245: Please expand FPGA.

The full form of FPGA, field-programmable gate arrays, is now added to **line 248** (please see the texts marked in red).

7. Line 246: Please define GUI.

The full form of GUI, graphical user interface, is now added to **line 250** (please see the texts marked in red).

8.Line 498: Please correct the typo “inducation”.

We thank the reviewer for pointing this out. The typo at **line 519** is now corrected to “induction” (please see the text marked in red).

References:

- 1 Servinsky, M. D. *et al.* Directed assembly of a bacterial quorum. *ISME J* **10**, 158-169, doi:10.1038/ismej.2015.89 (2016).
- 2 Bhokisham, N. *et al.* A redox-based electrogenetic CRISPR system to connect with and control biological information networks. *Nat Commun* **11**, 2427, doi:10.1038/s41467-020-16249-x (2020).
- 3 Li, J. *et al.* Mediated Electrochemistry to Mimic Biology's Oxidative Assembly of Functional Matrices. *Advanced Functional Materials* **30**, 2001776, doi:<https://doi.org/10.1002/adfm.202001776> (2020).
- 4 Joyner, K., Song, D., Hawkins, R. F., Silcott, R. D. & Duncan, G. A. A rational approach to form disulfide linked mucin hydrogels. *Soft Matter* **15**, 9632-9639, doi:10.1039/c9sm01715a (2019).
- 5 Sultana, S. T. *et al.* Electrochemical scaffold generates localized, low concentration of hydrogen peroxide that inhibits bacterial pathogens and biofilms. *Sci Rep* **5**, 14908, doi:10.1038/srep14908 (2015).
- 6 Ma, Z., Li, Z., Liu, K., Ye, C. & Sorger, V. J. Indium-Tin-Oxide for High-performance Electro-optic Modulation. *Nanophotonics* **4**, 198-213, doi:doi:10.1515/nanoph-2015-0006 (2015).

REVIEWERS' COMMENTS

Reviewer #1 (Remarks to the Author):

The authors have done a great job in addressing my concerns as well as those of the other reviewers. Congratulations.

Reviewer #2 (Remarks to the Author):

In the revised manuscript, the authors have made improvements and addressed most of the previous comments. However, a few concerns persist and should be addressed during revision.

1. What is the shelf life of devices like CRISPR modules in BioSpark? Could the authors demonstrate the longest shelf life of such devices at the current stage (Figure 6)?

2. Once assembled and used, would the devices be recycled or disposed of? Considering the inconsistent behavior of biofilms, please discuss how reproducibility will be maintained, and how the device output will be standardized if they need to be freshly assembled before each use.

3. For QS sensing signaling, the authors utilized an H₂O₂-inducible system. However, when *Escherichia coli* grows on standard substrates, it consistently generates 10 to 15 μ M of intracellular H₂O₂ through the inadvertent autoxidation of redox enzymes. Is there any impact of this endogenous H₂O₂ on the H₂O₂-inducible luxS CRISPRi and multiplexed control of QS signaling? The authors should discuss this issue.

4. The authors used 200 μ M of H₂O₂ to regulate autoinducer production. However, prior reports indicated that this concentration has some toxic effects and also induces various mutations in the genome (<https://doi.org/10.1371/journal.pgen.1008649>; <https://doi.org/10.1128/jb.169.7.2967-2976.1987>).

5. The authors should provide justifications for the H₂O₂ concentration chosen and discuss the issue.

6. While the H₂O₂-inducible system works well in this study, it might not be the most convenient and consistent. If applicable, please discuss and give examples of other signaling molecules that can be applied to these devices.

Reviewer #1 (Remarks to the Author):

Summary

The authors have done a great job in addressing my concerns as well as those of the other reviewers. Congratulations.

We are thankful for the reviewer's careful review of the manuscript and for the suggestions that have improved the manuscript.

Reviewer #2 (Remarks to the Author):

In the revised manuscript, the authors have made improvements and addressed most of the previous comments. However, a few concerns persist and should be addressed during revision.

We are thankful for the reviewer's careful review of the manuscript and for the suggestions that have improved the manuscript. Below are our responses to their specific comments:

1. What is the shelf life of devices like CRISPR modules in BioSpark? Could the authors demonstrate the longest shelf life of such devices at the current stage (Figure 6)?

We have not addressed the longevity of cells electroassembled into hydrogels. This will naturally depend on the availability of nutrients, water, and importantly the host chassis. This is actually a topic we are now exploring in a study aimed to develop actionable sensor films using *Pseudomonas*. For the current paper, we point to the longevity currently demonstrated, which is sufficient for the synthesis of proteins in a feedback-controlled manner. Specifically, in the original **Fig. 6** (now **Fig. 7**), the experiment ran for 8.5 h after deposition in BioSpark. We also showed the "growth profile" of up to 12 hours post-deposition for an artificial biofilm containing GFP-overexpressing *E. coli* (DH5 α -sfGFP). This is in **Supplementary Fig. 12a**.

2. Once assembled and used, would the devices be recycled or disposed of? Considering the inconsistent behavior of biofilms, please discuss how reproducibility will be maintained, and how the device output will be standardized if they need to be freshly assembled before each use.

The reviewer has touched on a very interesting avenue for further exploration. We envision both disposable and recyclable applications. We have developed sensor chips that "snap" into a microelectronic housing. These chips would be disposable, but not the housing. Alternatively, one might envision an application where the circuitry and communications modalities are expensive (a smart capsule for internal human use), and these would have surface exposed electrodes that would be reused. For the latter, we practiced this for the

current work. To ensure consistency, especially to maintain a clean working electrode surface, we carefully followed consistent deposition and washing protocols developed by us. These are in our **Methods** section. Interestingly, one can examine the measured current profile during electrodeposition to ensure the artificial biofilm is properly deposited. We found these are highly reproducible. Also, we found that by implementing these “standardized” protocols, our results (i.e., fluorescence raw values) were fairly consistent when using the same cells deposited for the same times, at the same pH and in identical solutions, and with the same OD. We would like to highlight that the data on controlled expression found in the new **Fig. 6** and in **Fig. 7**, are actually completely different experiments. They look amazingly similar and point to the reproducibility of the hydrogel assembly and, perhaps more importantly, the utility of on-line control. New text pertaining to the reproducibility of experiments using hydrogels can be found in the final manuscript under section “Electro-biofabrication: assembly of biological components onto electrodes.”

3. For QS sensing signaling, the authors utilized an H₂O₂-inducible system. However, when *Escherichia coli* grows on standard substrates, it consistently generates 10 to 15 μM of intracellular H₂O₂ through the inadvertent autoxidation of redox enzymes. Is there any impact of this endogenous H₂O₂ on the H₂O₂-inducible *luxS* CRISPRi and multiplexed control of QS signaling? The authors should discuss this issue.

We agree with the reviewer and acknowledge that *E. coli* produces endogenous peroxide during respiration. This is inevitable and reflects on the higher basal level of expression that is readily observed in our data. For example, in **Supplementary Fig. 7c**, NB101 cells harboring *luxS* eCRISPRi components exhibited lower AI-2 levels compared to the no plasmid control even when no peroxide or charge is applied for induction. We believe there were basal levels of gRNA expressed that were induced by endogenous gRNA. The same case (higher AI-1 and lower AI-2 levels without exogenous induction compared to no plasmid controls) can be observed in all CRISPR QS experiments. We tried to lower this basal expression by lowering plasmid copy numbers and adding degradation tags. We also showed in a previous work, that one can use CRISPRi to downregulate otherwise upregulated defense response genes in the *soxRS* regulon (another oxidative stress regulon) to amplify output and improve plasmid-encoded *soxS* promoter activity¹. Similar strategies could be implemented in future work to increase fold-change in peroxide-induced, CRISPR-mediated expression. Additionally, our next steps are towards different hosts that are more suited for other applications.

4. The authors used 200 μM of H₂O₂ to regulate autoinducer production. However, prior

reports indicated that this concentration has some toxic effects and also induces various mutations in the genome (<https://doi.org/10.1371/journal.pgen.1008649>; <https://doi.org/10.1128/jb.169.7.2967-2976.1987>).

We recognize that H₂O₂ can be a challenge, both in terms of toxicity and genotypic shifts. We hope the following data can address the reviewer's concern relative to cytotoxicity and potential mutations induced by exogenous peroxide induction. Our data in **Supplementary Fig. 4a (ii)** and **Supplementary Fig. 7d (iii)** depict the OD₆₀₀, or cell count, of the CRISPRa or multiplexed CRISPR cells induced by exogenous peroxide additions. In **Supplementary Fig. 4a (ii)**, we found significant statistical differences in OD₆₀₀ between the uninduced (0 μM) and induced (6.25-200 μM) cultures at hour 4. However, these differences are small (e.g., median OD₆₀₀ of the uninduced and 200 μM sample are 1.04 and 0.95, respectively) and we concluded peroxide (from 0-200 μM) exhibited minimal cytotoxicity to the eCRISPRa cells under the current experimental conditions. In **Supplementary Fig. 7d (iii)**, we found multiplexed CRISPR cells that received several rounds of 200 μM peroxide show only slightly lower OD₆₀₀ than cells that received none or one round of 200 μM peroxide, again suggesting peroxide exhibited minimal toxicity. Finally, in our previous work² albeit for different cells and conditions, we showed *E. coli* harboring plasmid-encoded *oxyR* and *oxyS* promoter all grew at similar rates post-induction irrespective of peroxide level until it reached above 500 μM.

In sum, one of the attributes afforded by the current design is that we can control the peroxide experienced by the cells. This is done electronically. We can dial down the amount of peroxide generated to that needed to minimally induce gene circuits. The amount of peroxide experienced by the cells at the surface of an electrode can be provided in small quantities and in pulses. These types of control situations cannot be done with chemically induced cells wherein one adds signaling molecules to various cell populations, including hydrogen peroxide.

5. The authors should provide justifications for the H₂O₂ concentration chosen and discuss the issue.

We have provided additional detail in the response to point #4. We also note parenthetically, the system works. We have shown real time feedback control using peroxide as an electronic signal. Cells are pulsed frequently; protein production is measured on-line and in real time. Then, in the end, a set point is reached. We expect miniaturized systems in which this level of production might be all that is required. We have added text in the **Discussion** that notes the peroxide issue and suggests further developments (CRISPRi, alternative hosts, etc.) for avoiding issues.

6. While the H₂O₂-inducible system works well in this study, it might not be the most convenient and consistent. If applicable, please discuss and give examples of other signaling molecules that can be applied to these devices.

We agree that there are a great many opportunities for electrogenetic actuation. In others, we add small-molecule redox mediators, and these carry the signal to the cells. For example, we previously generated another version of gene circuit for electrogenetic control that is based on another oxidative stress regulon *soxRS*^{1,3}. This system can be electrically controlled with redox-cycling compounds (e.g., pyocyanin, phenazine-methosulfate) paired with synthetic redox mediators (e.g., ferricyanide). We have also recently used a plant derived signal molecule, acetosyringone⁴ and showed how it can be used to activate gene expression using both *soxRS* and *oxyRS*. We would also like to note, however, that through QS message relay, the transmitter cells (after induced electrically via peroxide generation) can spread signaling molecules that are more stable and non-toxic (such as AI-1) to other nearby populations and coordinate their behavior⁵⁻⁷. This is actually our preferred avenue for future studies.

References:

- 1 Bhokisham, N. *et al.* A redox-based electrogenetic CRISPR system to connect with and control biological information networks. *Nat Commun* **11**, 2427, doi:10.1038/s41467-020-16249-x (2020).
- 2 Virgile, C. *et al.* Engineering bacterial motility towards hydrogen-peroxide. *PLoS One* **13**, e0196999, doi:10.1371/journal.pone.0196999 (2018).
- 3 Tschirhart, T. *et al.* Electronic control of gene expression and cell behaviour in *Escherichia coli* through redox signalling. *Nat Commun* **8**, 14030, doi:10.1038/ncomms14030 (2017).
- 4 Zakaria, F. R. *et al.* Redox Active Plant Phenolic, Acetosyringone, for Electrogenetic Signaling. *bioRxiv*, 2023.2009.2020.558642, doi:10.1101/2023.09.20.558642 (2023).
- 5 Terrell, J. L. *et al.* Bioelectronic control of a microbial community using surface-assembled electrogenetic cells to route signals. *Nat Nanotechnol* **16**, 688-697, doi:10.1038/s41565-021-00878-4 (2021).
- 6 VanArsdale, E. *et al.* Electrogenetic Signal Transmission and Propagation in Coculture to Guide Production of a Small Molecule, Tyrosine. *ACS Synth Biol* **11**, 877-887, doi:10.1021/acssynbio.1c00522 (2022).
- 7 VanArsdale, E. *et al.* Electrogenetic signaling and information propagation for controlling microbial consortia via programmed lysis. *Biotechnol Bioeng*, doi:10.1002/bit.28337 (2023).